# Error Bounds of Imitating Policies and Environments[*]

**Tian Xu[1], Ziniu Li[2,3], Yang Yu[1,3]**
[1]National Key Laboratory for Novel Software Technology,
Nanjing University, Nanjing 210023, China
[2]The Chinese University of Hong Kong, Shenzhen, Shenzhen 518172, China
[3]Polixir Technologies, Nanjing 210038, China
xut@lamda.nju.edu.cn, ziniuli@link.cuhk.edu.cn, yuy@nju.edu.cn

## Abstract

Imitation learning trains a policy by mimicking expert demonstrations. Various imitation methods were proposed and empirically evaluated, meanwhile, their theoretical understanding needs further studies. In this paper, we firstly analyze the value gap between the expert policy and imitated policies by two imitation methods, behavioral cloning and generative adversarial imitation. The results support that generative adversarial imitation can reduce the compounding errors compared to behavioral cloning, and thus has a better sample complexity. Noticed that by considering the environment transition model as a dual agent, imitation learning can also be used to learn the environment model. Therefore, based on the bounds of imitating policies, we further analyze the performance of imitating environments. The results show that environment models can be more effectively imitated by generative adversarial imitation than behavioral cloning, suggesting a novel application of adversarial imitation for model-based reinforcement learning. We hope these results could inspire future advances in imitation learning and model-based reinforcement learning.

## 1   Introduction

Sequential decision-making under uncertainty is challenging due to the stochastic dynamics and delayed feedback [27, 8]. Compared to reinforcement learning (RL) [46, 38] that learns from delayed feedback, imitation learning (IL) [37, 34, 23] learns from expert demonstrations that provide immediate feedback and thus is efficient in obtaining a good policy, which has been demonstrated in playing games [45], robotic control [19], autonomous driving [14], etc.

Imitation learning methods have been designed from various perspectives. For instance, behavioral cloning (BC) [37, 50] learns a policy by directly minimizing the action probability discrepancy with supervised learning; apprenticeship learning (AL) [1, 47] infers a reward function from expert demonstrations by inverse reinforcement learning [34], and subsequently learns a policy by reinforcement learning using the recovered reward function. Recently, Ho and Ermon [23] revealed that AL can be viewed as a state-action occupancy measure matching problem. From this connection, they proposed the algorithm generative adversarial imitation learning (GAIL). In GAIL, a discriminator scores the agent's behaviors based on the similarity to the expert demonstrations, and the agent learns to maximize the scores, resulting in expert-like behaviors.

Many empirical studies of imitation learning have been conducted. It has been observed that, for example, GAIL often achieves better performance than BC [23, 28, 29]. However, the theoretical

---

[*]This work is supported by National Key R&D Program of China (2018AAA0101100), NSFC (61876077), and Collaborative Innovation Center of Novel Software Technology and Industrialization. Yang Yu is the corresponding author. This work was done when Ziniu Li was an intern in Polixir Technologies.

explanations behind this observation have not been fully understood. Only until recently, there emerged studies towards understanding the generalization and computation properties of GAIL [13, 55]. In particular, Chen *et al.* [13] studied the generalization ability of the so-called $\mathcal{R}$-distance given the complete expert trajectories, while Zhang *et al.* [55] focused on the global convergence properties of GAIL under sophisticated neural network approximation assumptions.

In this paper, we present error bounds on the value gap between the expert policy and imitated policies from BC and GAIL, as well as the sample complexity of the methods. The error bounds indicate that the policy value gap is quadratic w.r.t. the horizon for BC, i.e., $1/(1 - \gamma)^2$, and cannot be improved in the worst case, which implies large compounding errors [39, 40]. Meanwhile, the policy value gap is only linear w.r.t. the horizon for GAIL, i.e., $1/(1 - \gamma)$. Similar to [13], the sample complexity also hints that controlling the complexity of the discriminator set in GAIL could be beneficial to the generalization. But our analysis strikes that a richer discriminator set is still required to reduce the policy value gap. Besides, our results provide theoretical support for the experimental observation that GAIL can generalize well even provided with incomplete trajectories [28].

Moreover, noticed that by regarding the environment transition model as a dual agent, imitation learning can also be applied to learn the transition model [51, 44, 43]. Therefore, based on the analysis of imitating policies, we further analyze the error bounds of imitating environments. The results indicate that the environment model learning through adversarial approaches enjoys a linear policy evaluation error w.r.t. the model-bias, which improves the previous quadratic results [31, 25] and suggests a promising application of GAIL for model-based reinforcement learning.

## 2 Background

### 2.1 Markov Decision Process

An infinite-horizon Markov decision process (MDP) [46, 38] is described by a tuple $\mathcal{M} = (\mathcal{S}, \mathcal{A}, M^*, R, \gamma, d_0)$, where $\mathcal{S}$ is the state space, $\mathcal{A}$ is the action space, and $d_0$ specifies the initial state distribution. We assume $\mathcal{S}$ and $\mathcal{A}$ are finite. The decision process runs as follows: at time step $t$, the agent observes a state $s_t$ from the environment and executes an action $a_t$, then the environment sends a reward $r(s_t, a_t)$ to the agent and transits to a new state $s_{t+1}$ according to $M^*(\cdot|s_t, a_t)$. Without loss of generality, we assume that the reward function is bounded by $R_{\max}$, i.e., $|r(s, a)| \le R_{\max}, \forall (s, a) \in \mathcal{S} \times \mathcal{A}$.

A (stationary) policy $\pi(\cdot|s)$ specifies an action distribution conditioned on state $s$. The quality of policy $\pi$ is evaluated by its policy value (i.e., cumulative rewards with a discount factor $\gamma \in [0, 1)$): $V_\pi = \mathbb{E}\left[\sum_{t=0}^{\infty} \gamma^t r(s_t, a_t)|s_0 \sim d_0, a_t \sim \pi(\cdot|s_t), s_{t+1} \sim M^*(\cdot|s_t, a_t), t = 0, 1, 2, \cdots\right]$. The goal of reinforcement learning is to find an optimal policy $\pi^*$ such that it maximizes the policy value (i.e., $\pi^* = \arg\max_\pi V_\pi$).

Notice that, in an infinite-horizon MDP, the policy value mainly depends on a finite length of the horizon due to the discount factor. The effective planning horizon [38] $1/(1 - \gamma)$, i.e., the total discounting weights of rewards, shows how the discount factor $\gamma$ relates with the effective horizon. We will see that the effective planning horizon plays an important role in error bounds of imitation learning approaches.

To facilitate later analysis, we introduce the *discounted stationary state distribution* $d_\pi(s) = (1 - \gamma)\sum_{t=0}^{\infty} \gamma^t \Pr(s_t = s; \pi)$, and the *discounted stationary state-action distribution* $\rho_\pi(s, a) = (1 - \gamma)\sum_{t=0}^{\infty} \gamma^t \Pr(s_t = s, a_t = a; \pi)$. Intuitively, discounted stationary state (state-action) distribution measures the overall "frequency" of visiting a state (state-action). For simplicity, we will omit "discounted stationary" throughout. We add a superscript to value function and state (state-action) distribution, e.g., $V_\pi^{M^*}$, when it is necessary to clarify the MDP.

### 2.2 Imitation Learning

Imitation learning (IL) [37, 34, 1, 47, 23] trains a policy by learning from expert demonstrations. In contrast to reinforcement learning, imitation learning is provided with action labels from expert policies. We use $\pi_E$ to denote the expert policy and $\Pi$ to denote the candidate policy class throughout this paper. In IL, we are interested in the policy value gap $V_{\pi_E} - V_\pi$. In the following, we briefly

introduce two popular methods considered in this paper, behavioral cloning (BC) [37] and generative adversarial imitation learning (GAIL) [23].

**Behavioral cloning.** In the simplest case, BC minimizes the action probability discrepancy with Kullback–Leibler (KL) divergence between the expert's action distribution and the imitating policy's action distribution. It can also be viewed as the maximum likelihood estimation in supervised learning.

$$\min_{\pi \in \Pi} \mathbb{E}_{s \sim d_{\pi_{\mathrm{E}}}} \left[ D_{\mathrm{KL}} \big( \pi_{\mathrm{E}}(\cdot|s), \pi(\cdot|s) \big) \right] := \mathbb{E}_{(s,a) \sim \rho_{\pi_{\mathrm{E}}}} \left[ \log \left( \frac{\pi_{\mathrm{E}}(a|s)}{\pi(a|s)} \right) \right]. \tag{1}$$

**Generative adversarial imitation learning.** In GAIL, a discriminator $D$ learns to recognize whether a state-action pair comes from the expert trajectories, while a generator $\pi$ mimics the expert policy via maximizing the rewards given by the discriminator. The optimization problem is defined as:

$$\min_{\pi \in \Pi} \max_{D \in (0,1)^{\mathcal{S} \times \mathcal{A}}} \mathbb{E}_{(s,a) \sim \rho_\pi} \left[ \log \big( D(s,a) \big) \right] + \mathbb{E}_{(s,a) \sim \rho_{\pi_{\mathrm{E}}}} \left[ \log (1 - D(s,a)) \right].$$

When the discriminator achieves its optimum $D^*(s,a) = \rho_\pi(s,a) / (\rho_\pi(s,a) + \rho_{\pi_{\mathrm{E}}}(s,a))$, we can derive that GAIL is to minimize the state-action distribution discrepancy between the expert policy and the imitating policy with the Jensen-Shannon (JS) divergence (up to a constant):

$$\min_{\pi \in \Pi} D_{\mathrm{JS}}(\rho_{\pi_{\mathrm{E}}}, \rho_\pi) := \frac{1}{2} \left[ D_{\mathrm{KL}}(\rho_\pi, \frac{\rho_\pi + \rho_{\pi_{\mathrm{E}}}}{2}) + D_{\mathrm{KL}}(\rho_{\pi_{\mathrm{E}}}, \frac{\rho_\pi + \rho_{\pi_{\mathrm{E}}}}{2}) \right]. \tag{2}$$

## 3 Related Work

In the domain of imitating policies, prior studies [39, 48, 40, 12] considered the finite-horizon setting and revealed that behavioral cloning [37] leads to the compounding errors (i.e., an optimality gap of $\mathcal{O}(T^2)$, where $T$ is the horizon length). DAgger [40] improved this optimality gap to $\mathcal{O}(T)$ at the cost of additional expert queries. Recently, based on generative adversarial network (GAN) [20], generative adversarial imitation learning [23] was proposed and had achieved much empirical success [17, 28, 29, 11]. Though many theoretical results have been established for GAN [5, 54, 3, 26], the theoretical properties of GAIL are not well understood. To the best of our knowledge, only until recently, there emerged studies towards understanding the generalization and computation properties of GAIL [13, 55]. The closest work to ours is [13], where the authors considered the generalization ability of GAIL under a finite-horizon setting with complete expert trajectories. In particular, they analyzed the generalization ability of the proposed $\mathcal{R}$-distance but they did not provide the bound for policy value gap, which is of interest in practice. On the other hand, the global convergence properties with neural network function approximation were further analyzed in [55].

In addition to BC and GAIL, apprenticeship learning (AL) [1, 47, 49, 24] is a promising candidate for imitation learning. AL infers a reward function from expert demonstrations by inverse reinforcement learning (IRL) [34], and subsequently learns a policy by reinforcement learning using the recovered reward function. In particular, IRL aims to identify a reward function under which the expert policy's performance is optimal. Feature expectation matching (FEM) [1] and game-theoretic apprenticeship learning (GTAL) [47] are two popular AL algorithms with theoretical guarantees under tabular MDP. To obtain an $\epsilon$-optimal policy, FEM and GTAL require expert trajectories of $\mathcal{O}(\frac{k \log k}{(1-\gamma)^2 \epsilon^2})$ and $\mathcal{O}(\frac{\log k}{(1-\gamma)^2 \epsilon^2})$ respectively, where $k$ is the number of predefined feature functions.

In addition to imitating policies, learning environment transition models can also be treated by imitation learning by considering environment transition model as a dual agent. This connection has been utilized in [44, 43] to model real-world environments and in [51] to reduce the regret regarding model-bias following the idea of DAgger [40], where the model-bias refers to prediction errors when a learned environment model predicts the next state given the current state and current action. Many studies [6, 31, 25] have shown that if the virtual environment is trained with the principle of behavioral cloning (i.e., minimizing one-step transition prediction errors), the learned policy from this learned environment also suffers from the issue of compounding errors regarding model-bias. We are also inspired by the connection between imitation learning and environment-learning, but we focus on applying the distribution matching property of generative adversarial imitation learning to alleviate model-bias. Noticed that in [44, 43], adversarial approaches have been adopted to learn a high fidelity virtual environment, but the reason why such approaches work was unclear. This paper provides a partial answer.

# 4 Bounds on Imitating Policies

## 4.1 Imitating Policies with Behavioral Cloning

It is intuitive to understand why behavioral cloning suffers from large compounding errors [48, 40] as that the imitated policy, even with a small training error, may visit a state out of the expert demonstrations, which causes a larger decision error and a transition to further unseen states. Consequently, the policy value gap accumulates along with the planning horizon. The error bound of BC has been established in [48, 40] under a finite-horizon setting, and here we present an extension to the infinite-horizon setting.

**Theorem 1.** *Given an expert policy $\pi_{\mathrm{E}}$ and an imitated policy $\pi_{\mathrm{I}}$ with $\mathbb{E}_{s \sim d_{\pi_{\mathrm{E}}}}[D_{\mathrm{KL}}(\pi_{\mathrm{E}}(\cdot|s), \pi_{\mathrm{I}}(\cdot|s))] \leq \epsilon$ (which can be achieved by BC with objective Eq.(1)), we have that $V_{\pi_{\mathrm{E}}} - V_{\pi_{\mathrm{I}}} \leq \frac{2\sqrt{2}R_{\max}}{(1-\gamma)^2}\sqrt{\epsilon}$.*

The proof by the coherent error-propagation analysis can be found in Appendix A. Note that Theorem 1 is under the infinite sample situation. In the finite sample situation, one can further bound the generalization error $\epsilon$ in the RHS using classical learning theory (see Corollary 1) and the proof can be found in Appendix A.

**Corollary 1.** *We use $\{(s_{\pi_{\mathrm{E}}}^{(i)}, a_{\pi_{\mathrm{E}}}^{(i)})\}_{i=1}^m$ to denote the expert demonstrations. Suppose that $\pi_{\mathrm{E}}$ and $\pi_{\mathrm{I}}$ are deterministic and the provided function class $\Pi$ satisfies realizability, meaning that $\pi_{\mathrm{E}} \in \Pi$. For policy $\pi_{\mathrm{I}}$ imitated by BC (see Eq. (1)), $\forall \delta \in (0,1)$, with probability at least $1 - \delta$, we have that*

$$V_{\pi_{\mathrm{E}}} - V_{\pi_{\mathrm{I}}} \leq \frac{2R_{\max}}{(1-\gamma)^2}\left(\frac{1}{m}\log(|\Pi|) + \frac{1}{m}\log\left(\frac{1}{\delta}\right)\right).$$

Moreover, we show that the value gap bound in Theorem 1 is tight up to a constant by providing an example shown in Figure 1 (more details can be found in Appendix A.3). Therefore, we conclude that the quadratic dependency on the effective planning horizon, $\mathcal{O}(1/(1-\gamma)^2)$, is inevitable in the worst-case.

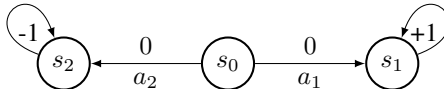

Figure 1: A "hard" deterministic MDP corresponding to Theorem 1. Digits on arrows are corresponding rewards. Initial state is $s_0$ while $s_1$ and $s_2$ are two absorbing states.

## 4.2 Imitating Policies with GAIL

Different from BC, GAIL [23] is to minimize the state-action distribution discrepancy with JS divergence. The state-action distribution discrepancy captures the temporal structure of Markov decision process, thus it is more favorable in imitation learning. Recent researches [35, 18] showed that besides JS divergence, discrepancy measures based on a general class, $f$-divergence [30, 16], can be applied to design discriminators. Given two distributions $\mu$ and $\nu$, $f$-divergence is defined as $D_f(\mu, \nu) = \int \mu(x) f(\frac{\mu(x)}{\nu(x)}) dx$, where $f(\cdot)$ is a convex function that satisfies $f(1) = 0$. Here, we consider GAIL using some common $f$-divergences listed in Table 1 in Appendix B.1.

**Lemma 1.** *Given an expert policy $\pi_{\mathrm{E}}$ and an imitated policy $\pi_{\mathrm{I}}$ with $D_f(\rho_{\pi_{\mathrm{I}}}, \rho_{\pi_{\mathrm{E}}}) \leq \epsilon$ (which can be achieved by GAIL) using the $f$-divergence in Table 1, we have that $V_{\pi_{\mathrm{E}}} - V_{\pi_{\mathrm{I}}} \leq \mathcal{O}(\frac{1}{1-\gamma}\sqrt{\epsilon})$.*

The proof can be found in Appendix B.1. Lemma 1 indicates that the optimality gap of GAIL grows linearly with the effective horizon $1/(1-\gamma)$, multiplied by the square root of the $f$-divergence error $D_f$. Compared to Theorem 1, this result indicates that GAIL with the $f$-divergence could have fewer compounding errors if the objective function is properly optimized. Note that this result does not claim that GAIL is overall better than BC, but can highlight that GAIL has a linear dependency on the planning horizon compared to the quadratic one in BC.

Analyzing the generalization ability of GAIL with function approximation is somewhat more complicated, since GAIL involves a minimax optimization problem. Most of the existing learning theories [32, 42], however, focus on the problems that train one model to minimize the empirical loss, and therefore are hard to be directly applied. In particular, the discriminator in GAIL is often parameterized by certain neural networks, and therefore it may not be optimum within a restrictive function class. In that case, we may view the imitated policy is to minimize the *neural network distance* [5] instead of the ideal $f$-divergence.

**Definition 1** (Neural network distance [5]). *For a class of neural networks $\mathcal{D}$, the neural network distance between two (state-action) distributions, $\mu$ and $\nu$, is defined as*

$$d_{\mathcal{D}}(\mu, \nu) = \sup_{D \in \mathcal{D}} \left\{ \mathbb{E}_{(s,a) \sim \mu}[D(s,a)] - \mathbb{E}_{(s,a) \sim \nu}[D(s,a)] \right\}.$$

Interestingly, it has been shown that the generalization ability of neural network distance is substantially different from the original divergence measure [5, 54] due to the limited representation ability of the discriminator set $\mathcal{D}$. For instance, JS-divergence may not generalize even with sufficient samples [5]. In the following, we firstly discuss the generalization ability of the neural network distance, based on which we formally give the upper bound of the policy value gap.

To ensure the non-negativity of neural network distance, we assume that the function class $\mathcal{D}$ contains the zero function, i.e., $\exists D \in \mathcal{D}, D(s,a) \equiv 0$. Neural network distance is also known as integral probability metrics (IPM) [33].

As an illustration, $f$-divergence is connected with neural network distance by its variational representation [54]:

$$d_{f,\mathcal{D}}(\mu, \nu) = \sup_{d \in \mathcal{D}} \left\{ \mathbb{E}_{(s,a) \sim \mu}[D(s,a)] - \mathbb{E}_{(s,a) \sim \nu}[D(s,a)] - \mathbb{E}_{(s,a) \sim \mu}[\phi^*(f(s,a))] \right\},$$

where $\phi^*$ is the (shifted) convex conjugate of $f$. Thus, considering $\phi^* = 0$ and choosing the activation function of the last layer in the discriminator as the sigmoid function $g(t) = 1/(1 + \exp(-t))$ recovers the original GAIL objective [54]. Again, such defined neural network distance is still different from the original $f$-divergence because of the limited representation ability of $\mathcal{D}$. Thereafter, we may consider GAIL is to find a policy $\pi_{\mathrm{I}}$ by minimizing $d_{\mathcal{D}}(\rho_{\pi_{\mathrm{E}}}, \rho_{\pi_{\mathrm{I}}})$.

As another illustration, when $\mathcal{D}$ is the class of all 1-Lipschitz continuous functions, $d_D(\mu, \nu)$ is the well-known Wasserstein distance [4]. From this viewpoint, we give an instance called Wasserstein GAIL (WGAIL) in Appendix D, where the discriminator in practice is to approximate 1-Lipschitz functions with neural networks. However, note that neural network distance in WGAIL is still distinguished from Wasserstein distance since $\mathcal{D}$ cannot contain all 1-Lipschitz continuous functions.

In practice, GAIL minimizes the empirical neural network distance $d_{\mathcal{D}}(\hat{\rho}_{\pi_{\mathrm{E}}}, \hat{\rho}_{\pi})$, where $\hat{\rho}_{\pi_{\mathrm{E}}}$ and $\hat{\rho}_{\pi}$ denote the empirical version of population distribution $\rho_{\pi_{\mathrm{E}}}$ and $\rho_{\pi}$ with $m$ samples. To analyze its generalization property, we employ the standard Rademacher complexity technique. The Rademacher random variable $\sigma$ is defined as $\Pr(\sigma = +1) = \Pr(\sigma = -1) = 1/2$. Given a function class $\mathcal{F}$ and a dataset $Z = (z_1, z_2, \cdots, z_m)$ that is i.i.d. drawn from distribution $\mu$, the empirical Rademacher complexity $\hat{\mathcal{R}}_{\mu}^{(m)}(\mathcal{F}) = \mathbb{E}_{\boldsymbol{\sigma}}[\sup_{f \in \mathcal{F}} \frac{1}{m} \sum_{i=1}^{m} \sigma_i f(z_i)]$ measures the richness of function class $\mathcal{F}$ by the ability to fit random variables [32, 42]. The generalization ability of GAIL is analyzed in [13] under a different definition. They focused on how many trajectories, rather than our focus on state-action pairs, are sufficient to guarantee generalization. Importantly, we further disclose the policy value gap in Theorem 2 based on the neural network distance.

**Lemma 2** (Generalization of neural network distance). *Consider a discriminator class set $\mathcal{D}$ with $\Delta$-bounded value functions, i.e., $|D(s,a)| \leq \Delta$, for all $(s,a) \in \mathcal{S} \times \mathcal{A}, D \in \mathcal{D}$. Given an expert policy $\pi_{\mathrm{E}}$ and an imitated policy $\pi_{\mathrm{I}}$ with $d_{\mathcal{D}}(\hat{\rho}_{\pi_{\mathrm{E}}}, \hat{\rho}_{\pi_{\mathrm{I}}}) - \inf_{\pi \in \Pi} d_{\mathcal{D}}(\hat{\rho}_{\pi_{\mathrm{E}}}, \hat{\rho}_{\pi}) \leq \hat{\epsilon}$, then $\forall \delta \in (0,1)$, with probability at least $1 - \delta$, we have*

$$d_{\mathcal{D}}(\rho_{\pi_{\mathrm{E}}}, \rho_{\pi_{\mathrm{I}}}) \leq \underbrace{\inf_{\pi \in \Pi} d_{\mathcal{D}}(\hat{\rho}_{\pi_{\mathrm{E}}}, \hat{\rho}_{\pi})}_{\mathrm{Appr}(\Pi)} + \underbrace{2\hat{\mathcal{R}}_{\rho_{\pi_{\mathrm{E}}}}^{(m)}(\mathcal{D}) + 2\hat{\mathcal{R}}_{\rho_{\pi_{\mathrm{I}}}}^{(m)}(\mathcal{D}) + 12\Delta\sqrt{\frac{\log(2/\delta)}{m}}}_{\mathrm{Estm}(\mathcal{D}, m, \delta)} + \hat{\epsilon}.$$

The proof can be found in Appendix B.3. Here $\mathrm{Appr}(\Pi)$ corresponds to the approximation error induced by the limited policy class $\Pi$. $\mathrm{Estm}(\mathcal{D}, m, \delta)$ denotes the estimation error of GAIL regarding

to the complexity of discriminator class and the number of samples. Lemma 2 implies that GAIL generalizes if the complexity of discriminator class $\mathcal{D}$ is properly controlled. Concretely, a simpler discriminator class reduces the estimation error, then tends to reduce the neural network distance. Here we provide an example of neural networks with ReLU activation functions to illustrate this.

**Example 1** (Neural Network Discriminator Class). We consider the neural networks with ReLU activation functions $(\sigma_1, \ldots, \sigma_L)$. We use $b_\mathrm{s}$ to denote the spectral norm bound and $b_\mathrm{n}$ to denote the matrix $(2, 1)$ norm bound. The discriminator class consists of $L$-layer neural networks $D_\mathbf{A}$:

$$\mathcal{D} := \left\{ D_\mathbf{A} : \mathbf{A} = (A_1, \ldots, A_L), \|A_i\|_\sigma \leq b_\mathrm{s}, \|A_i^\top\|_{2,1} \leq b_\mathrm{n}, \forall i \in \{1, \ldots, L\} \right\},$$

where $D_\mathbf{A}(s, a) = \sigma_L(A_L \cdots \sigma_1(A_1[s^\top, a^\top]^\top))$. Then the spectral normalized complexity $R_\mathbf{A}$ of network $D_\mathbf{A}$ is $\mathcal{O}(L^{\frac{3}{2}} b_\mathrm{s}^{L-1} b_\mathrm{n})$ (see [9] for more details). Derived by the Theorem 3.4 in [9] and Lemma 2, with probability at least $1 - \delta$, we have

$$d_\mathcal{D}(\rho_{\pi_\mathrm{E}}, \rho_{\pi_\mathrm{I}}) \leq \mathcal{O}\left( \frac{L^{\frac{3}{2}} b_\mathrm{s}^{L-1} b_\mathrm{n}}{m} \left( 1 + \log\left( \frac{m}{L^{\frac{3}{2}} b_\mathrm{s}^{L-1} b_\mathrm{n}} \right) \right) + \Delta\sqrt{\frac{\log(1/\delta)}{m}} \right) + \inf_{\pi \in \Pi} d_\mathcal{D}(\hat{\rho}_{\pi_\mathrm{E}}, \hat{\rho}_\pi) + \hat{\epsilon}.$$

From a theoretical view, reducing the number of layers $L$ could reduce the spectral normalized complexity $R_\mathbf{A}$ and the neural network distance $d_\mathcal{D}(\rho_{\pi_\mathrm{E}}, \rho_{\pi_\mathrm{I}})$. However, we did not empirically observe that this operation significantly affects the performance of GAIL. On the other hand, consistent with [28, 29], we find the gradient penalty technique [21] can effectively control the model complexity since this technique gives preference for 1-Lipschitz continuous functions. In this way, the number of candidate functions decreases, and thus the Rademacher complexity of discriminator class is controlled. We also note that the information bottleneck [2] technique helps to control the model complexity and to improve GAIL's performance in practice [36] but the rigorous theoretical explanation is unknown.

However, when the discriminator class is restricted to a set of neural networks with relatively small complexity, it is not safe to conclude that the policy value gap $V_{\pi_\mathrm{E}} - V_{\pi_\mathrm{I}}$ is small when $d_\mathcal{D}(\rho_{\pi_\mathrm{E}}, \rho_{\pi_\mathrm{I}})$ is small. As an extreme case, if the function class $\mathcal{D}$ only contains constant functions, the neural network distance always equals to zero while the policy value gap could be large. Therefore, we still need a richer discriminator set to distinguish different policies. To substantiate this idea, we introduce the linear span of the discriminator class: $\mathrm{span}(\mathcal{D}) = \{c_0 + \sum_{i=1}^n c_i D_i : c_0, c_i \in \mathbb{R}, D_i \in \mathcal{D}, n \in \mathbb{N}\}$. Furthermore, we assume that $\mathrm{span}(\mathcal{D})$, rather than $\mathcal{D}$, has enough capacity such that the ground truth reward function $r$ lies in it and define the *compatible coefficient* as:

$$\|r\|_\mathcal{D} = \inf \left\{ \sum_{i=1}^n |c_i| : r = \sum_{i=1}^n c_i D_i + c_0, \forall n \in \mathbb{N}, c_0, c_i \in \mathbb{R}, D_i \in \mathcal{D} \right\}.$$

Here, $\|r\|_\mathcal{D}$ measures the minimum number of functions in $\mathcal{D}$ required to represent $r$ and $\|r\|_\mathcal{D}$ decreases when the discriminator class becomes richer. Now we present the result on generalization ability of GAIL in the view of policy value gap.

**Theorem 2** (GAIL Generalization). *Under the same assumption of Lemma 2 and suppose that the ground truth reward function $r$ lies in the linear span of discriminator class, with probability at least $1 - \delta$, the following inequality holds.*

$$V_{\pi_\mathrm{E}} - V_{\pi_\mathrm{I}} \leq \frac{\|r\|_\mathcal{D}}{1 - \gamma} \big( \mathrm{Appr}(\Pi) + \mathrm{Estm}(\mathcal{D}, m, \delta) + \hat{\epsilon} \big).$$

The proof can be found in Appendix B.4. Theorem 2 discloses that the policy value gap grows linearly with the effective horizon, due to the global structure of state-action distribution matching. This is an advantage of GAIL when only a few expert demonstrations are provided. Moreover, Theorem 2 suggests seeking a trade-off on the complexity of discriminator class: a simpler discriminator class enjoys a smaller estimation error, but could enlarge the compatible coefficient. Finally, Theorem 2 implies the generalization ability holds when provided with some state-action pairs, explaining the phenomenon that GAIL still performs well when only having access to incomplete trajectories [28]. One of the limitations of our result is that we do not deeply consider the approximation ability of the policy class and its computation properties with stochastic policy gradient descent.

# 5 Bounds on Imitating Environments

The task of environment learning is to recover the transition model of MDP, $M_\theta$, from data collected in the real environment $M^*$, and it is the core of model-based reinforcement learning (MBRL). Although environment learning is typically a separated topic with imitation learning, it has been noticed that learning environment transition model can also be treated by imitation learning [51, 44, 43]. In particular, a transition model takes the state $s$ and action $a$ as input and predicts the next state $s'$, which can be considered as a dual agent, so that the imitation learning can be applied. The learned transition probability $M_\theta(s'|s, a)$ is expected to be close to the true probability $M^*(s'|s, a)$. Under the background of MBRL, we assess the quality of the learned transition model $M_\theta$ by the evaluation error of an arbitrary policy $\pi$, i.e., $|V_\pi^{M^*} - V_\pi^{M_\theta}|$, where $V_\pi^{M^*}$ is the true value and $V_\pi^{M_\theta}$ is the value in the learned transition model. Note that we focus on learning the transition model, while assuming the true reward function is always available[2]. For simplicity, we only present the error bounds here and it's feasible to extend our results with the concentration measures to obtain finite sample complexity bounds.

## 5.1 Imitating Environments with Behavioral Cloning

Similarly, we can directly employ behavioral cloning to minimize the one-step prediction errors when imitating environments, which is formulated as the following optimization problem.

$$\min_\theta \mathbb{E}_{(s,a)\sim\rho_{\pi_D}^{M^*}} \left[ D_{\mathrm{KL}}\big(M^*(\cdot|s,a), M_\theta(\cdot|s,a)\big) \right] := \mathbb{E}_{(s,a)\sim\rho_{\pi_D}^{M^*}, s'\sim M^*(\cdot|s,a)} \left[ \log \frac{M^*(s'|s,a)}{M_\theta(s'|s,a)} \right],$$

where $\pi_D$ denotes the data-collecting policy and $\rho_{\pi_D}^{M^*}$ denotes its state-action distribution. We will see that the issue of compounding errors also exists in model-based policy evaluation. Intuitively, if the learned environment cannot capture the transition model globally, the policy evaluation error blows up regarding the model-bias, which degenerates the effectiveness of MBRL. In the following, we formally state this result for self-containing, though similar results have been appeared in [31, 25].

**Lemma 3.** *Given a true MDP with transition model $M^*$, a data-collecting policy $\pi_D$, and a learned transition model $M_\theta$ with $\mathbb{E}_{(s,a)\sim\rho_{\pi_D}^{M^*}} \left[ D_{\mathrm{KL}}\big(M^*(\cdot|s,a), M_\theta(\cdot|s,a)\big) \right] \le \epsilon_m$, for an arbitrary bounded divergence policy $\pi$, i.e., $\max_s D_{\mathrm{KL}}\big(\pi(\cdot|s), \pi_D(\cdot|s)\big) \le \epsilon_\pi$, the policy evaluation error is bounded by $|V_\pi^{M^*} - V_\pi^{M_\theta}| \le \frac{\sqrt{2}R_{\max}\gamma}{(1-\gamma)^2}\sqrt{\epsilon_m} + \frac{2\sqrt{2}R_{\max}}{(1-\gamma)^2}\sqrt{\epsilon_\pi}.$*

Note that the policy evaluation error contains two terms, the inaccuracy of the learned model measured under the state-action distribution of the data-collecting policy $\pi_D$, and the policy divergence between $\pi$ and $\pi_D$. We realize that the $1/(1-\gamma)^2$ dependency on the policy divergence $\epsilon_\pi$ is inevitable (see also the Theorem 1 in TRPO [41]). Hence, we mainly focus on how to reduce the model-bias term.

## 5.2 Imitating environments with GAIL

As shown in Lemma 1, GAIL mitigates the issue of compounding errors via matching the state-action distribution of expert policies. Inspired by this observation, we analyze the error bound of GAIL for environment-learning tasks. Concretely, we train a transition model (also as a policy) that takes state $s_t$ and action $a_t$ as inputs and outputs a distribution over next state $s_{t+1}$; at the meantime, we also train a discriminator that learns to recognize whether a state-action-next-state tuple $(s_t, a_t, s_{t+1})$ comes from the "expert" demonstrations, where the "expert" demonstrations should be explained as the transitions collected by running the data-collecting policy in the true environment. This procedure is summarized in Algorithm 1 in Appendix C.3. It is easy to verify that all the occupancy measure properties of GAIL for imitating policies are reserved by 1) augmenting the action space into the new state space; 2) treating the next state space as the action space when imitating environments. In the following, we show that, by GAIL, the dependency on the effective horizon is only linear in the term of model-bias.

We denote $\mu^{M_\theta}$ as the state-action-next-state distribution of the data-collecting policy $\pi_D$ in the learned transition model, i.e., $\mu^{M_\theta}(s, a, s') = M_\theta(s'|s, a)\rho_{\pi_D}^{M_\theta}(s, a)$; and $\mu^{M^*}$ as that in the true transition model. The proof of Theorem 3 can be found in Appendix C.

**Theorem 3.** *Given a true MDP with transition model $M^*$, a data-collecting policy $\pi_D$, and a learned transition model $M_\theta$ with $D_{\mathrm{JS}}(\mu^{M_\theta}, \mu^{M^*}) \leq \epsilon_m$, for an arbitrary bounded divergence policy $\pi$, i.e. $\max_s D_{\mathrm{KL}}\big(\pi(\cdot|s), \pi_D(\cdot|s)\big) \leq \epsilon_\pi$, the policy evaluation error is bounded by $|V_\pi^{M_\theta} - V_\pi^{M^*}| \leq \frac{2\sqrt{2}R_{\max}}{1-\gamma}\sqrt{\epsilon_m} + \frac{2\sqrt{2}R_{\max}}{(1-\gamma)^2}\sqrt{\epsilon_\pi}$.*

Theorem 3 suggests that recovering the environment transition with a GAIL-style learner can mitigate the model-bias when evaluating policies. We provide the experimental evidence in Section 6.2. Combing this model-imitation technique with all kinds of policy optimization algorithms is an interesting direction that we will explore in the future.

# 6 Experiments

## 6.1 Imitating Policies

We evaluate imitation learning methods on three MuJoCo benchmark tasks in OpenAI Gym [10], where the agent aims to mimic locomotion skills. We consider the following approaches: BC [37], DAgger [40], GAIL [23], maximum entropy IRL algorithm AIRL [17] and apprenticeship learning algorithms FEM [1] and GTAL [47]. In particular, FEM and GTAL are based on the improved versions proposed in [24]. Besides GAIL, we also involve WGAIL (see Appendix D) in the comparisons. We run the state-of-the-art algorithm SAC [22] to obtain expert policies. All experiments run with 3 random seeds. Experiment details are given in Appendix E.1.

**Study of effective horizon dependency.** We firstly compare the methods with different effective planning horizons. All approaches are provided with only 3 expert trajectories, except for DAgger that continues to query expert policies during training. When expert demonstrations are scanty, the impact of the scaling factor $1/(1 - \gamma)$ could be significant. The relative performance (i.e., $V_\pi/V_{\pi_{\mathrm{E}}}$) of learned policies under MDPs with different discount factors $\gamma$ is plotted in Figure 2. Note that the performance of expert policies increases as the planning horizon increases, thus the decrease trends of some curves do not imply the decrease trends of policy values. Exact results and learning curves are given in Appendix E. Though some polices may occasionally outperform experts in short planning horizon settings, we care mostly whether a policy can match the performance of expert policies when the planning horizon increases. We can see that when the planning horizon increases, BC is worse than GAIL, and possibly AIRL, FEM and GTAL. The observation confirms our analysis. Consistent with [23], we also have empirically observed that when BC is provided lots of expert demonstrations, the training error and generalization error could be very small. In that case, the discount scaling factor does not dominate and BC's performance is competitive.

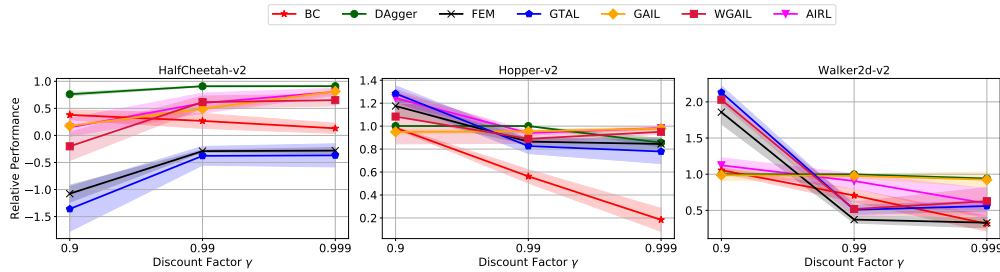

Figure 2: Relative performance of imitated policies under MDPs with different discount factors $\gamma$.

**Study of generalization ability of GAIL.** We then empirically validate the trade-off about the model complexity of the discriminator set in GAIL. We realize that neural networks used by the discriminator are often over-parameterized on MuJoCo tasks and find that carefully using the gradient penalty technique [21] can control the model's complexity to obtain better generalization results. In particular, gradient penalty incurs a quadratic cost function to the gradient norm, which makes the discriminator set a preference to 1-Lipschitz continuous functions. This loss function is multiplied by a coefficient of $\lambda$ and added to the original objective function. Learning curves of varying $\lambda$ are given in Figure 3.

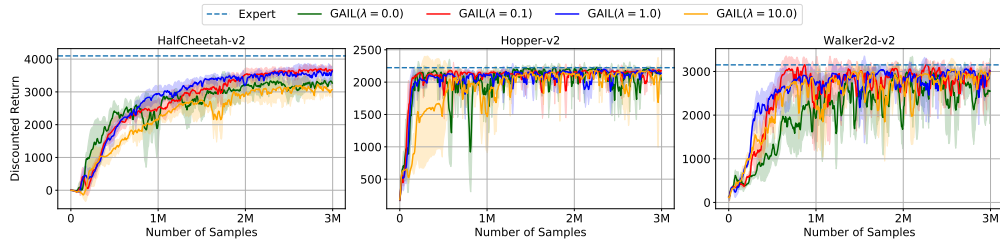

Figure 3: Learning curves of GAIL ($\gamma = 0.999$) with different gradient penalty coefficients $\lambda$.

We can see that a moderate $\lambda$ (e.g., $0.1$ or $1.0$) yields better performance than a large $\lambda$ (e.g., $10$) or small $\lambda$ (e.g., $0$).

## 6.2 Imitating Environments

We conduct experiments to verify that generative adversarial learning could mitigate the model-bias for imitating environments in the setting of model-based reinforcement learning. Here we only focus on the comparisons between GAIL and BC for environment-learning. Both methods are provided with 20 trajectories to learn the transition model. Experiment details are given in Appendix E.2. We evaluate the performance by the policy evaluation error $|V_{\pi_D}^{M_\theta} - V_{\pi_D}^{M^*}|$, where $\pi_D$ denotes the data-collecting policy. As we can see in Figure 4, the policy evaluation errors are smaller on all three environments learned by GAIL. Note that BC tends to over-fit, thus the policy evaluation errors do not decrease on HalfCheetah-v2.

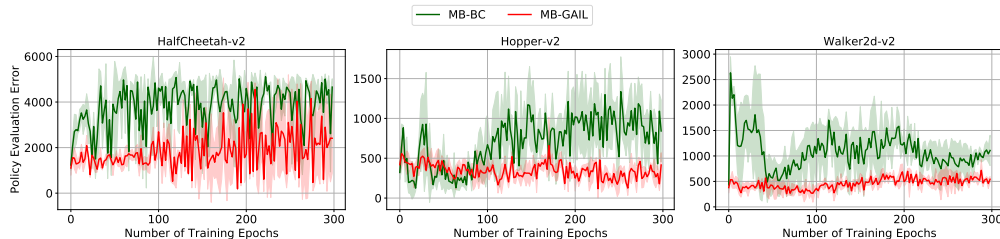

Figure 4: Policy evaluation errors ($\gamma = 0.999$) on environment models trained by BC and GAIL.

## 7 Conclusion

This paper presents error bounds of BC and GAIL for imitating-policies and imitating-environments in the infinite horizon setting, mainly showing that GAIL can achieve a linear dependency on the effective horizon while BC has a quadratic dependency. The results can enhance our understanding of imitation learning methods.

We would like to highlight that the result of the paper may shed some light for model-based reinforcement learning (MBRL). Previous MBRL methods mostly involve a BC-like transition learning component that can cause a high model-bias. Our analysis suggests that the BC-like transition learner can be replaced by a GAIL-style learner to improve the generalization ability, which also partially addresses the reason that why GAIL-style environment model learning approach in [44, 43] can work well. Learning a useful environment model is an essential way towards sample-efficient reinforcement learning [52], which is not only because the environment model can directly be used for cheap training, but it is also an important support for meta-reinforcement learning (e.g., [53]). We hope this work will inspire future research in this direction.

Our analysis of GAIL focused on the generalization ability of the discriminator and further analysis of the computation and approximation ability of the policy was left for future works.

## Broader Impact

This work focuses on the theoretical understanding about imitation learning methods in imitating policies and environments, which does not present any direct societal consequence. This work indicates possible improvement direction for MBRL, which might help reinforcement learning get better used in the real world. There could be some consequence when reinforcement learning is getting abused, such as manipulate information presentation to control people's behaviors.

## Acknowledgments and Disclosure of Funding

The authors would like to thank Dr. Weinan Zhang and Dr. Zongzhang Zhang for their helpful comments. This work is supported by National Key R&D Program of China (2018AAA0101100), NSFC (61876077), and Collaborative Innovation Center of Novel Software Technology and Industrialization.

## Footnotes

[2]In the case where the reward function is unknown, we can directly learn the reward function with supervised learning and the corresponding sample complexity is a lower order term compared to the one of learning the transition model [7].

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
