[Supplementary Material]

# A Analysis of Imitating-policies with BC

Here, we present an error propagation analysis to derive the compounding errors of BC under the setting of infinite-horizon MDP. Our derivation is based on the framework of error-propagation (see Figure 5), which illustrates the cause of compounding errors. Note that the error-propagation framework focuses on the absolute value of policy value gap $|V_\pi - V_{\pi_E}|$, and the one side bound $V_\pi - V_{\pi_E}$ can be easily derived from it.

Figure 5: Error propagation of behavioral cloning.

## A.1 Error-propagation Analysis

We firstly introduce the following Lemma, which tells that how much state distribution discrepancy grows based on the policy distribution discrepancy.

**Lemma 4.** *For two policies $\pi$ and $\pi_E$, we have that*

$$D_{\mathrm{TV}}(d_\pi, d_{\pi_E}) \leq \frac{\gamma}{1-\gamma} \mathbb{E}_{s \sim d_{\pi_E}} \left[ D_{\mathrm{TV}}\big(\pi(\cdot|s), \pi_E(\cdot|s)\big) \right].$$

*Proof.* The proof is based on the permutation theory presented in [41]. First, we show that

$$
\begin{aligned}
d_\pi &= (1-\gamma) \sum_{t=0}^{\infty} \gamma^t \Pr(s_t = s|\pi, d_0) \\
&= (1-\gamma)(I - \gamma P_\pi)^{-1} d_0,
\end{aligned}
$$

where $P_\pi(s'|s) = \sum_{a \in A} M^*(s'|s, a)\pi(a|s)$. Then we obtain that

$$
\begin{aligned}
d_\pi - d_{\pi_E} &= (1-\gamma)[(I - \gamma P_\pi)^{-1} - (I - \gamma P_{\pi_E})^{-1}] \, d_0 \\
&= (1-\gamma)(M_\pi - M_{\pi_E}) \, d_0,
\end{aligned}
\tag{3}
$$

where $M_\pi = (I - \gamma P_\pi)^{-1}$ and $M_{\pi_E} = (I - \gamma P_{\pi_E})^{-1}$. For the term $M_\pi - M_{\pi_E}$, we obtain that

$$
\begin{aligned}
M_\pi - M_{\pi_E} &= M_\pi \left( M_{\pi_E}^{-1} - M_\pi^{-1} \right) M_{\pi_E} \\
&= \gamma M_\pi (P_\pi - P_{\pi_E}) M_{\pi_E}.
\end{aligned}
\tag{4}
$$

Combining Eq. (3) with Eq. (4), we have

$$
\begin{aligned}
d_\pi - d_{\pi_E} &= (1-\gamma)\gamma M_\pi (P_\pi - P_{\pi_E}) M_{\pi_E} d_0 \\
&= \gamma M_\pi (P_\pi - P_{\pi_E}) d_{\pi_E}.
\end{aligned}
$$

Therefore, we obtain that

$$D_{\mathrm{TV}}(d_\pi, d_{\pi_{\mathrm{E}}}) = \frac{\gamma}{2}\|M_\pi(P_\pi - P_{\pi_{\mathrm{E}}})d_{\pi_{\mathrm{E}}}\|_1$$

$$\leq \frac{\gamma}{2}\|M_\pi\|_1\|(P_\pi - P_{\pi_{\mathrm{E}}})d_{\pi_{\mathrm{E}}}\|_1. \tag{5}$$

We can show that $M_\pi$ is bounded:

$$\|M_\pi\|_1 = \|\sum_{t=0}^{\infty}\gamma^t P_\pi^t\|_1 \leq \sum_{t=0}^{\infty}\gamma^t\|P_\pi\|_1^t \leq \sum_{t=0}^{\infty}\gamma^t = \frac{1}{1-\gamma}.$$

Consequently, we show that $\|(P_\pi - P_{\pi_{\mathrm{E}}})d_{\pi_{\mathrm{E}}}\|_1$ is also bounded,

$$\|(P_\pi - P_{\pi_{\mathrm{E}}})d_{\pi_{\mathrm{E}}}\|_1 \leq \sum_{s,s'}|P_\pi(s'|s) - P_{\pi_{\mathrm{E}}}(s'|s)|\,d_{\pi_{\mathrm{E}}}(s)$$

$$= \sum_{s,s'}\left|\sum_a M^*(s'|s,a)\big(\pi(a|s) - \pi_{\mathrm{E}}(a|s)\big)\right|d_{\pi_{\mathrm{E}}}(s)$$

$$\leq \sum_{(s,a),s'}M^*(s'|s,a)|\pi(a|s) - \pi_{\mathrm{E}}(a|s)|d_{\pi_{\mathrm{E}}}(s)$$

$$= \sum_s d_{\pi_{\mathrm{E}}}(s)\sum_a|\pi(a|s) - \pi_{\mathrm{E}}(a|s)|$$

$$= 2\mathbb{E}_{s\sim d_{\pi_{\mathrm{E}}}}[D_{\mathrm{TV}}\big(\pi_{\mathrm{E}}(\cdot|s), \pi(\cdot|s)\big)].$$

Combining Eq. (5) with the above two inequalities completes the proof. □

Next, we further bound the state-action distribution discrepancy based on the policy discrepancy.

**Lemma 5.** *For any two policies $\pi$ and $\pi_{\mathrm{E}}$, we have that*

$$D_{\mathrm{TV}}(\rho_\pi, \rho_{\pi_{\mathrm{E}}}) \leq \frac{1}{1-\gamma}\mathbb{E}_{s\sim d_{\pi_{\mathrm{E}}}}\left[D_{\mathrm{TV}}\big(\pi(\cdot|s), \pi_{\mathrm{E}}(\cdot|s)\big)\right].$$

*Proof.* Note that the relationship $\rho_\pi(s,a) = \pi(a|s)d_\pi(s)$ for any policy $\pi$, we have

$$D_{\mathrm{TV}}(\rho_\pi, \rho_{\pi_{\mathrm{E}}})$$

$$= \frac{1}{2}\sum_{(s,a)}\left|\big[\pi_{\mathrm{E}}(a|s) - \pi(a|s)\big]d_{\pi_{\mathrm{E}}}(s) + \big[d_{\pi_{\mathrm{E}}}(s) - d_\pi(s)\big]\pi(a|s)\right|$$

$$\leq \frac{1}{2}\sum_{(s,a)}|\pi_{\mathrm{E}}(a|s) - \pi(a|s)|d_{\pi_{\mathrm{E}}}(s) + \frac{1}{2}\sum_{(s,a)}\pi(a|s)|d_{\pi_{\mathrm{E}}}(s) - d_\pi(s)|$$

$$= \mathbb{E}_{s\sim d_{\pi_{\mathrm{E}}}}[D_{\mathrm{TV}}\big(\pi(\cdot|s), \pi_{\mathrm{E}}(\cdot|s)\big)] + D_{\mathrm{TV}}(d_\pi, d_{\pi_{\mathrm{E}}})$$

$$\leq \frac{1}{1-\gamma}\mathbb{E}_{s\sim d_{\pi_{\mathrm{E}}}}[D_{\mathrm{TV}}\big(\pi(\cdot|s), \pi_{\mathrm{E}}(\cdot|s)\big)],$$

where the last inequality follows Lemma 4. □

Finally, we bound the policy value gap (i.e., the difference between value of learned policy $\pi$ and the expert policy $\pi_{\mathrm{E}}$) based on the state-action distribution discrepancy.

**Lemma 6.** *For any two policies $\pi$ and $\pi_{\mathrm{E}}$, we have that*

$$|V_\pi - V_{\pi_{\mathrm{E}}}| \leq \frac{2R_{\max}}{1-\gamma}D_{\mathrm{TV}}(\rho_\pi, \rho_{\pi_{\mathrm{E}}}).$$

*Proof.* It is a well-known fact that for any policy $\pi$, its policy value can be reformulated as $V^\pi = \frac{1}{1-\gamma}\mathbb{E}_{(s,a)\sim\rho_\pi}[r(s,a)]$ [38]. Based on this observation, we derive that

$$
\begin{aligned}
|V_\pi - V_{\pi_\mathrm{E}}| &= \left| \frac{1}{1-\gamma}\mathbb{E}_{(s,a)\sim\rho_\pi}[r(s,a)] - \frac{1}{1-\gamma}\mathbb{E}_{(s,a)\sim\rho_{\pi_\mathrm{E}}}[r(s,a)] \right| \\
&\leq \frac{1}{1-\gamma}\sum_{(s,a)\in\mathcal{S}\times\mathcal{A}} \left| \big(\rho_\pi(s,a) - \rho_{\pi_\mathrm{E}}(s,a)\big)r(s,a) \right| \\
&\leq \frac{2R_{\max}}{1-\gamma}D_{\mathrm{TV}}(\rho_\pi, \rho_{\pi_\mathrm{E}}).
\end{aligned}
$$

$\square$

## A.2 Proof of Theorem 1

*Proof of Theorem 1.* Suppose that the imitated policy $\pi_\mathrm{I}$ optimizes the objective of BC up to an $\epsilon$ error, i.e., $\mathbb{E}_{s\sim d_{\pi_\mathrm{E}}}\big[D_{\mathrm{KL}}\big(\pi_\mathrm{I}(\cdot|s), \pi_\mathrm{E}(\cdot|s)\big)\big] \leq \epsilon$. Combining Lemma 5 and Lemma 6, we have that, for policy $\pi_\mathrm{I}$ and $\pi_\mathrm{E}$,

$$
\begin{aligned}
V_{\pi_\mathrm{E}} - V_{\pi_\mathrm{I}} &\leq \frac{2R_{\max}}{1-\gamma}D_{\mathrm{TV}}(\rho_{\pi_\mathrm{I}}, \rho_{\pi_\mathrm{E}}) \\
&\leq \frac{2R_{\max}}{(1-\gamma)^2}\mathbb{E}_{s\sim d_{\pi_\mathrm{E}}}\big[D_{\mathrm{TV}}\big(\pi_\mathrm{I}(\cdot|s), \pi_\mathrm{E}(\cdot|s)\big)\big].
\end{aligned}
$$

Thanks to Pinsker's inequality [15] that for two arbitrary distributions $\mu$ and $\nu$, $D_{\mathrm{TV}}(\mu,\nu) \leq \sqrt{2D_{\mathrm{KL}}(\mu,\nu)}$, we obtain that

$$
\begin{aligned}
V_{\pi_\mathrm{E}} - V_{\pi_\mathrm{I}} &\leq \frac{2R_{\max}}{(1-\gamma)^2}\mathbb{E}_{s\sim d_{\pi_\mathrm{E}}}\left[\sqrt{2D_{\mathrm{KL}}\big(\pi_\mathrm{I}(\cdot|s), \pi_\mathrm{E}(\cdot|s)\big)}\right] \\
&\leq \frac{2\sqrt{2}R_{\max}}{(1-\gamma)^2}\sqrt{\mathbb{E}_{s\sim d_{\pi_\mathrm{E}}}\big[D_{\mathrm{KL}}\big(\pi_\mathrm{I}(\cdot|s), \pi_\mathrm{E}(\cdot|s)\big)\big]} \\
&\leq \frac{2\sqrt{2}R_{\max}}{(1-\gamma)^2}\sqrt{\epsilon_\pi},
\end{aligned}
$$

where the penultimate inequality follows Jensen's inequality $\phi(\mathbb{E}[X]) \leq \mathbb{E}[\phi(X)]$, where $\phi(x) = -\sqrt{x}$.

$\square$

Based on Theorem 1, we provide a sample complexity analysis of BC using classical learning theory.

*Proof of Corollary 1.* From Lemma 5 and Lemma 6, we obtain that

$$
V_{\pi_\mathrm{E}} - V_{\pi_\mathrm{I}} \leq \frac{2R_{\max}}{(1-\gamma)^2}\mathbb{E}_{s\sim d_{\pi_\mathrm{E}}}\big[D_{\mathrm{TV}}\big(\pi_\mathrm{I}(\cdot|s), \pi_\mathrm{E}(\cdot|s)\big)\big]. \tag{6}
$$

Here we consider that $\pi_\mathrm{I}$ and $\pi_\mathrm{E}$ are deterministic policies, thus we obtain that

$$
\mathbb{E}_{s\sim d_{\pi_\mathrm{E}}}[D_{\mathrm{TV}}(\pi(\cdot|s), \pi_\mathrm{E}(\cdot|s))] = \mathbb{E}_{s\sim d_{\pi_\mathrm{E}}}[\mathbb{I}(\pi(s) \neq \pi_\mathrm{E}(s))],
$$

where $\mathbb{I}$ is the indicator function. The policy $\pi_\mathrm{I}$ is obtained by solving Eq.(1), thus $\pi_\mathrm{I}\big(s_{\pi_\mathrm{E}}^{(i)}\big) = a_{\pi_\mathrm{E}}^{(i)}, \forall i \in \{1, \cdots, m\}$. Since behavioral cloning employs supervised learning to learn a policy, we follow the standard argument in the classical learning theory [32] in the remaining proof. We define the expected risk $L(\pi) = \mathbb{E}_{s\sim d_{\pi_\mathrm{E}}}[\mathbb{I}(\pi(s) \neq \pi_\mathrm{E}(s))]$ and the empirical risk $L_m(\pi) = \frac{1}{m}\sum_{i=1}^m \mathbb{I}(\pi(s_{\pi_\mathrm{E}}^{(i)}) \neq a_{\pi_\mathrm{E}}^{(i)})$. For a fixed $\epsilon > 0$, we define the bad policy class $\Pi_\mathrm{B} = \{\pi \in \Pi : L(\pi) > \epsilon\}$. Then we bound the probability of policy $\pi_\mathrm{I}$ belongs to the bad policy class $\Pi_\mathrm{B}$:

$$
\Pr(L(\pi_\mathrm{I}) > \epsilon) = \Pr(\pi_\mathrm{I} \in \Pi_\mathrm{B}).
$$

Because the empirical risk of $\pi_\mathrm{I}$ equals zero, we get that

$$
\Pr(\pi_\mathrm{I} \in \Pi_\mathrm{B}) \leq \Pr(\exists \pi \in \Pi, L_m(\pi) = 0).
$$

For a fixed $\pi \in \Pi$, $\Pr(L_m(\pi) = 0) = (1 - L(\pi))^m \leq (1 - \epsilon)^m \leq e^{-\epsilon m}$, where the last step follows $1 - a \leq e^{-a}$. Then we obtain that

$$\Pr(L(\pi_{\mathrm{I}}) > \epsilon) \leq \Pr(\exists \pi \in \Pi, L_m(\pi) = 0) \leq \sum_{\pi \in \Pi_{\mathrm{B}}} \Pr(L_m(\pi) = 0) \leq |\Pi| e^{-\epsilon m}.$$

Setting the right-hand side to be equal to $\delta$, we get that $L(\pi_{\mathrm{I}}) \leq \frac{1}{m} \left( \log(|\Pi|) + \log(\frac{1}{\delta}) \right)$. Combining it with Eq. (6) completes the proof. $\qquad\square$

### A.3 Tightness of Theorem 1

Figure 6: A "hard" deterministic MDP corresponding to Theorem 1. Digits on arrows are corresponding rewards. Initial state is $s_0$ while $s_1$ and $s_2$ are two absorbing states.

Here we validate that the $\gamma$-dependence in Theorem 1 is tight by the simple example in Figure 6. Note that the initial state is $s_0$ and two absorbing states are $s_1$ and $s_2$. That is, the agent always starts with $s_0$ and takes an action $a_1$ ($a_2$); consequently, the system transits into the absorbing state $s_1$ ($s_2$). Here we consider a sub-optimal expert policy $\pi_{\mathrm{E}}$ that chooses $a_1$ with probability of 0.9 and chooses $a_2$ with probability of 0.1 at $s_0$, meaning that $\pi_{\mathrm{E}}(a_1|s_0) = 0.9$, $\pi_{\mathrm{E}}(a_2|s_0) = 0.1$ and we can show that the policy value of expert policy $\pi_{\mathrm{E}}$ is $V_{\pi_{\mathrm{E}}} = \frac{4\gamma}{5(1-\gamma)}$. In addition, we can show that the state distribution of expert policy $\pi_{\mathrm{E}}$ is $d_{\pi_{\mathrm{E}}} = (d_{\pi_{\mathrm{E}}}(s_0), d_{\pi_{\mathrm{E}}}(s_1), d_{\pi_{\mathrm{E}}}(s_2)) = (1 - \gamma, \frac{9}{10}\gamma, \frac{1}{10}\gamma)$. Consider a policy obtained by behavioral cloning $\pi_{\mathrm{I}}$ that chooses $a_1$ at $s_0$ with probability of 0.85 and $a_2$ with probability of 0.15, meaning that $\pi_{\mathrm{I}}(a_1|s_0) = 0.85, \pi_{\mathrm{I}}(a_2|s_0) = 0.15$. Similarly, we can show that $V_{\pi_{\mathrm{I}}} = \frac{7\gamma}{10(1-\gamma)}$ and the policy value gap $V_{\pi_{\mathrm{E}}} - V_{\pi_{\mathrm{I}}} = \frac{\gamma}{10(1-\gamma)}$. It is easy to verify that the error bound $\mathbb{E}_{s \sim d_{\pi_{\mathrm{E}}}} \left[ D_{\mathrm{KL}}\left( \pi_{\mathrm{E}}(\cdot|s), \pi(\cdot|s) \right) \right]$ on the RHS of Eq. (1) is about $0.011(1 - \gamma)$ and consequently $V_{\pi_{\mathrm{E}}} - V_{\pi_{\mathrm{I}}} = C \cdot \frac{1}{(1-\gamma)^2} \mathbb{E}_{s \sim d_{\pi_{\mathrm{E}}}} \left[ D_{\mathrm{KL}}\left( \pi_{\mathrm{E}}(\cdot|s), \pi(\cdot|s) \right) \right]$, where $C$ is a constant. The equality implies that in the worst case, the quadratic discount complexity is tight in Theorem 1.

## B  Analysis of Imitating-policies with GAIL

### B.1  $f$-divergence

A large class of divergence measures called $f$-divergence [30] can be applied to depict the difference between two probability distributions. Given two probability density function $\mu$ and $\nu$ with respect to a base measure defined on the domain $\mathcal{X}$, $f$-divergence is defined as

$$D_f(\mu, \nu) = \int_{\mathcal{X}} \mu(x) f\left(\frac{\mu(x)}{\nu(x)}\right) dx,$$

where $f(\cdot)$ is a convex function that satisfies $f(1) = 0$. Different choices of $f$ decides specific measures. When $f(u) = -(u + 1) \log(\frac{1+u}{2}) + u \log(u)$, $f$-divergence recovers the JS divergence used in GAIL. Table 1 lists many of the common $f$-divergences and the $f$ functions to which they correspond (see also [35]). In the following, we provide a proof of Lemma 1. The proof is based on the concentration between different $f$-divergences.

### B.2  Proof of Lemma 1

*Proof of Lemma 1.* Here we prove that GAIL with $f$-divergence listed in Table 1 enjoys a linear policy value gap. Derived from Lemma 6, we obtain that

$$V_{\pi_{\mathrm{E}}} - V_{\pi} \leq \frac{2R_{\max}}{1 - \gamma} D_{\mathrm{TV}}(\rho_{\pi}, \rho_{\pi_{\mathrm{E}}}). \tag{7}$$

Table 1: List of $f$-divergences

| Name | $D_f(\mu, \nu)$ | $f(u)$ |
|---|---|---|
| Kullback-Leibler | $\int \mu(x) \log(\frac{\mu(x)}{\nu(x)}) dx$ | $u \log(u)$ |
| Reverse KL | $\int \nu(x) \log(\frac{\nu(x)}{\mu(x)}) dx$ | $-\log(u)$ |
| Pearsion $\chi^2$ | $\int \frac{(\mu(x)-\nu(x))^2}{\mu(x)} dx$ | $(u-1)^2$ |
| Jensen-Shannon | $\frac{1}{2} \int \mu(x) \log(\frac{2\mu(x)}{\mu(x)+\nu(x)}) + \nu(x) \log(\frac{2\nu(x)}{\mu(x)+\nu(x)}) dx$ | $-(u+1) \log(\frac{u+1}{2}) + u \log(u)$ |
| Squared Hellinger | $\int (\sqrt{\mu(x)} - \sqrt{\nu(x)})^2 dx$ | $(\sqrt{u} - 1)^2$ |

**JS divergence:**
In the following, we connect the total variation with the JS divergence based on Pinsker's inequality,

$$
\begin{aligned}
D_{\mathrm{JS}}(\rho_{\pi_{\mathrm{I}}}, \rho_{\pi_{\mathrm{E}}}) &= \frac{1}{2} \left( D_{\mathrm{KL}}(\rho_{\pi_{\mathrm{I}}}, \frac{\rho_{\pi_{\mathrm{I}}} + \rho_{\pi_{\mathrm{E}}}}{2}) + D_{\mathrm{KL}}(\rho_{\pi_{\mathrm{E}}}, \frac{\rho_{\pi_{\mathrm{I}}} + \rho_{\pi_{\mathrm{E}}}}{2}) \right) \\
&\geq D_{\mathrm{TV}}^2(\rho_{\pi_{\mathrm{I}}}, \frac{\rho_{\pi_{\mathrm{I}}} + \rho_{\pi_{\mathrm{E}}}}{2}) + D_{\mathrm{TV}}^2(\rho_{\pi_{\mathrm{E}}}, \frac{\rho_{\pi_{\mathrm{I}}} + \rho_{\pi_{\mathrm{E}}}}{2}) \\
&= \frac{1}{2} D_{\mathrm{TV}}^2(\rho_{\pi_{\mathrm{I}}}, \rho_{\pi_{\mathrm{E}}}).
\end{aligned}
\tag{8}
$$

Combining Eq. (7) with Eq. (8), we get that

$$
V_{\pi_{\mathrm{E}}} - V_{\pi_{\mathrm{I}}} \leq \frac{2\sqrt{2} R_{\max}}{1 - \gamma} \sqrt{D_{\mathrm{JS}}(\rho_{\pi_{\mathrm{I}}}, \rho_{\pi_{\mathrm{E}}})}.
$$

**KL divergence & Reverse KL divergence:**
Again, thanks to Pinsker's inequality, we obtain that the policy value gap is bounded by KL divergence and Reverse KL divergence.

$$
V_{\pi_{\mathrm{E}}} - V_{\pi_{\mathrm{I}}} \leq \frac{\sqrt{2} R_{\max}}{1 - \gamma} \sqrt{D_{\mathrm{KL}}(\rho_{\pi_{\mathrm{I}}}, \rho_{\pi_{\mathrm{E}}})}.
$$

$$
V_{\pi_{\mathrm{E}}} - V_{\pi_{\mathrm{I}}} \leq \frac{\sqrt{2} R_{\max}}{1 - \gamma} \sqrt{D_{\mathrm{KL}}(\rho_{\pi_{\mathrm{E}}}, \rho_{\pi_{\mathrm{I}}})}.
$$

For $\chi^2$ divergence and Squared Hellinger divergence, we can build similar upper bounds of policy value gap.

$$
V_{\pi_{\mathrm{E}}} - V_{\pi_{\mathrm{I}}} \leq \frac{R_{\max}}{1 - \gamma} \sqrt{\chi^2(\rho_{\pi_{\mathrm{I}}}, \rho_{\pi_{\mathrm{E}}})}
$$

$$
V_{\pi_{\mathrm{E}}} - V_{\pi_{\mathrm{I}}} \leq \frac{2 R_{\max}}{1 - \gamma} \sqrt{D_{\mathrm{H}}(\rho_{\pi_{\mathrm{I}}}, \rho_{\pi_{\mathrm{E}}})}.
$$

In conclusion, for policy $\pi_{\mathrm{I}}$ imitated by GAIL with $f$-divergence listed in Table 1, we have that $V_{\pi_{\mathrm{E}}} - V_{\pi_{\mathrm{I}}} \leq \mathcal{O}\left(\frac{1}{1-\gamma} \sqrt{D_f(\rho_{\pi_{\mathrm{I}}}, \rho_{\pi_{\mathrm{E}}})}\right)$, which finishes the proof. $\qquad \square$

### B.3 Proof of Lemma 2

*Proof of Lemma 2.* When the policy $\pi_{\mathrm{I}}$ optimizes the empirical GAIL loss $d_{\mathcal{D}}(\hat{\rho}_{\pi_{\mathrm{E}}}, \hat{\rho}_{\pi})$ up to an $\epsilon_{\mathrm{opt}}$ error, we have that

$$
d_{\mathcal{D}}(\hat{\rho}_{\pi_{\mathrm{E}}}, \hat{\rho}_{\pi_{\mathrm{I}}}) \leq \inf_{\pi \in \Pi} d_{\mathcal{D}}(\hat{\rho}_{\pi_{\mathrm{E}}}, \hat{\rho}_{\pi}) + \epsilon_{\mathrm{opt}},
\tag{9}
$$

where $\hat{\rho}_{\pi_{\mathrm{E}}}$ denotes the expert demonstrations with $m$ state-action pairs $\{(s_{\pi_{\mathrm{E}}}^{(i)}, a_{\pi_{\mathrm{E}}}^{(i)})\}_{i=1}^{m}$ and $\hat{\rho}_{\pi_{\mathrm{I}}}$ is the empirical version of population distribution $\rho_{\pi_{\mathrm{I}}}$ with $m$ samples $\{(s_{\pi_{\mathrm{I}}}^{(i)}, a_{\pi_{\mathrm{I}}}^{(i)})\}_{i=1}^{m}$ collected by $\pi_{\mathrm{I}}$. By standard derivation, we get that

$$
d_{\mathcal{D}}(\rho_{\pi_{\mathrm{E}}}, \rho_{\pi_{\mathrm{I}}}) \leq d_{\mathcal{D}}(\rho_{\pi_{\mathrm{E}}}, \rho_{\pi_{\mathrm{I}}}) - d_{\mathcal{D}}(\hat{\rho}_{\pi_{\mathrm{E}}}, \hat{\rho}_{\pi_{\mathrm{I}}}) + \inf_{\pi \in \Pi} d_{\mathcal{D}}(\hat{\rho}_{\pi_{\mathrm{E}}}, \hat{\rho}_{\pi}) + \epsilon_{\mathrm{opt}}.
\tag{10}
$$

According to the definition of neural network distance $d_{\mathcal{D}}(\mu, \nu)$, we prove that $d_{\mathcal{D}}(\rho_{\pi_E}, \rho_{\pi_I}) - d_{\mathcal{D}}(\hat{\rho}_{\pi_E}, \hat{\rho}_{\pi_I})$ has an upper bound.

$$
\begin{aligned}
& d_{\mathcal{D}}(\rho_{\pi_E}, \rho_{\pi_I}) - d_{\mathcal{D}}(\hat{\rho}_{\pi_E}, \hat{\rho}_{\pi_I}) \\
&= \sup_{D \in \mathcal{D}} \left[ \mathbb{E}_{(s,a) \sim \rho_{\pi_E}}[D(s,a)] - \mathbb{E}_{(s,a) \sim \rho_{\pi_I}}[D(s,a)] \right] - \sup_{D \in \mathcal{D}} \left[ \mathbb{E}_{(s,a) \sim \hat{\rho}_{\pi_E}}[D(s,a)] - \mathbb{E}_{(s,a) \sim \hat{\rho}_{\pi_I}}[D(s,a)] \right] \\
&\leq \sup_{D \in \mathcal{D}} \left\{ \left[ \mathbb{E}_{(s,a) \sim \rho_{\pi_E}}[D(s,a)] - \mathbb{E}_{(s,a) \sim \rho_{\pi_I}}[D(s,a)] \right] - \left[ \mathbb{E}_{(s,a) \sim \hat{\rho}_{\pi_E}}[D(s,a)] - \mathbb{E}_{(s,a) \sim \hat{\rho}_{\pi_I}}[D(s,a)] \right] \right\} \\
&\leq \sup_{D \in \mathcal{D}} \left[ \mathbb{E}_{(s,a) \sim \rho_{\pi_E}}[D(s,a)] - \mathbb{E}_{(s,a) \sim \hat{\rho}_{\pi_E}}[D(s,a)] \right] + \sup_{D \in \mathcal{D}} \left[ \mathbb{E}_{(s,a) \sim \hat{\rho}_{\pi_I}}[D(s,a)] - \mathbb{E}_{(s,a) \sim \rho_{\pi_I}}[D(s,a)] \right] \\
&\leq \sup_{D \in \mathcal{D}} \left| \mathbb{E}_{(s,a) \sim \rho_{\pi_E}}[D(s,a)] - \mathbb{E}_{(s,a) \sim \hat{\rho}_{\pi_E}}[D(s,a)] \right| + \sup_{D \in \mathcal{D}} \left| \mathbb{E}_{(s,a) \sim \rho_{\pi_I}}[D(s,a)] - \mathbb{E}_{(s,a) \sim \hat{\rho}_{\pi_I}}[D(s,a)] \right|.
\end{aligned}
$$

We first show that $\sup_{D \in \mathcal{D}} \left| \mathbb{E}_{(s,a) \sim \rho_{\pi_E}}[D(s,a)] - \mathbb{E}_{(s,a) \sim \hat{\rho}_{\pi_E}}[D(s,a)] \right|$ can be bounded. Note that the assumption that the discriminator set $\mathcal{D}$ consists of bounded functions with $\Delta$, i.e. $\sup_{D \in \mathcal{D}} \|D(s,a)\|_\infty \leq \Delta, \forall (s,a) \in \mathcal{S} \times \mathcal{A}$. According to McDiarmid 's inequality [32], with probability at least $1 - \frac{\delta}{4}$, the following inequality holds.

$$
\begin{aligned}
& \sup_{D \in \mathcal{D}} \left| \mathbb{E}_{(s,a) \sim \rho_{\pi_E}}[D(s,a)] - \mathbb{E}_{(s,a) \sim \hat{\rho}_{\pi_E}}[D(s,a)] \right| \\
& \leq \mathbb{E} \left[ \sup_{D \in \mathcal{D}} \left| \mathbb{E}_{(s,a) \sim \rho_{\pi_E}}[D(s,a)] - \mathbb{E}_{(s,a) \sim \hat{\rho}_{\pi_E}}[D(s,a)] \right| \right] + 2\Delta \sqrt{\frac{\log(4/\delta)}{2m}},
\end{aligned}
\tag{11}
$$

where the outer expectation is taken over the random choice of expert demonstrations $\hat{\rho}_{\pi_E}$ with $m$ state-action pairs. According to the Rademacher complexity theory [32], for the first term of Eq. (11) we have that

$$
\begin{aligned}
& \mathbb{E} \left[ \sup_{D \in \mathcal{D}} \left| \mathbb{E}_{(s,a) \sim \rho_{\pi_E}}[D(s,a)] - \mathbb{E}_{(s,a) \sim \hat{\rho}_{\pi_E}}[D(s,a)] \right| \right] \\
& \leq 2 \mathbb{E}_{\boldsymbol{\sigma}, \rho_{\pi_E}} \left[ \sup_{D \in \mathcal{D}} \sum_{i=1}^{m} \frac{1}{m} \sigma_i D(s^{(i)}, a^{(i)}) \right] \\
& = 2 \mathcal{R}_{\rho_{\pi_E}}^{(m)}(\mathcal{D}).
\end{aligned}
\tag{12}
$$

Based on the connection between Rademacher complexity and empirical Rademacher complexity, we have that with probability at least $1 - \frac{\delta}{4}$, the following inequality holds.

$$
\mathcal{R}_{\rho_{\pi_E}}^{(m)}(\mathcal{D}) \leq \hat{\mathcal{R}}_{\rho_{\pi_E}}^{(m)}(\mathcal{D}) + 2\Delta \sqrt{\frac{\log(4/\delta)}{2m}},
\tag{13}
$$

where $\hat{\mathcal{R}}_{\rho_{\pi_E}}^{(m)}(\mathcal{D}) = \mathbb{E}_{\boldsymbol{\sigma}} \left[ \sup_{D \in \mathcal{D}} \sum_{i=1}^{m} \frac{1}{m} \sigma_i D(s_{\pi_E}^{(i)}, a_{\pi_E}^{(i)}) \right]$. Combining Eq. (11) with Eq. (13), with probability at least $1 - \frac{\delta}{2}$, we have

$$
\sup_{D \in \mathcal{D}} \left| \mathbb{E}_{(s,a) \sim \rho_{\pi_E}}[D(s,a)] - \mathbb{E}_{(s,a) \sim \hat{\rho}_{\pi_E}}[D(s,a)] \right| \leq 2 \hat{\mathcal{R}}_{\rho_{\pi_E}}^{(m)}(\mathcal{D}) + 6\Delta \sqrt{\frac{\log(4/\delta)}{2m}}.
\tag{14}
$$

By a similar derivation, we obtain that with probability at least $1 - \frac{\delta}{2}$, the following inequality holds.

$$
\sup_{D \in \mathcal{D}} \left| \mathbb{E}_{(s,a) \sim \rho_{\pi_I}}[D(s,a)] - \mathbb{E}_{(s,a) \sim \hat{\rho}_{\pi_I}}[D(s,a)] \right| \leq 2 \hat{\mathcal{R}}_{\rho_{\pi_I}}^{(m)}(\mathcal{D}) + 6\Delta \sqrt{\frac{\log(4/\delta)}{2m}},
\tag{15}
$$

where $\hat{\mathcal{R}}_{\rho_{\pi_I}}^{(m)}(\mathcal{D}) = \mathbb{E}_{\boldsymbol{\sigma}} \left[ \sup_{D \in \mathcal{D}} \sum_{i=1}^{m} \frac{1}{m} \sigma_i D(s_{\pi_I}^{(i)}, a_{\pi_I}^{(i)}) \right]$. Combining Eq. (10) with Eq. (14) and Eq. (15), we complete the proof. $\qquad \square$

### B.4 Proof of Theorem 2

*Proof of Theorem 2.* We use the re-formulation of policy value $V_\pi = \frac{1}{1-\gamma} \mathbb{E}_{(s,a) \sim \rho_\pi}[r(s,a)]$ and derive that

$$
V_{\pi_E} - V_{\pi_I} \leq \frac{1}{1-\gamma} \left| \mathbb{E}_{(s,a) \sim \rho_{\pi_I}}[r(s,a)] - \mathbb{E}_{(s,a) \sim \rho_{\pi_E}}[r(s,a)] \right|.
$$

As we assume that the reward function $r$ lies in the linear span of $\mathcal{D}$, there exists $n \in \mathbb{N}$, $\{c_i \in \mathbb{R}\}_{i=1}^n$ and $\{D_i \in \mathcal{D}\}_{i=1}^n$, such that $r = c_0 + \sum_{i=1}^n c_i D_i$. Noticed by $c_0$ will be eliminated by the difference of policy value, we obtain that

$$
\begin{aligned}
V_{\pi_{\mathrm{E}}} - V_{\pi_{\mathrm{I}}} &\leq \frac{1}{1-\gamma} \left| \sum_{i=1}^n c_i \mathbb{E}_{(s,a)\sim\rho_{\pi_{\mathrm{I}}}}[D_i(s,a)] - \sum_{i=1}^n c_i \mathbb{E}_{(s,a)\sim\rho_{\pi_{\mathrm{E}}}}[D_i(s,a)] \right| \\
&\leq \frac{1}{1-\gamma} \sum_{i=1}^n |c_i| \left| \mathbb{E}_{(s,a)\sim\rho_{\pi_{\mathrm{I}}}}[D_i(s,a)] - \mathbb{E}_{(s,a)\sim\rho_{\pi_{\mathrm{E}}}}[D_i(s,a)] \right| \\
&\leq \frac{1}{1-\gamma} \left( \sum_{i=1}^n |c_i| \right) d_{\mathcal{D}}(\rho_{\pi_{\mathrm{I}}}, \rho_{\pi_{\mathrm{E}}}) \\
&\leq \frac{1}{1-\gamma} \|r\|_{\mathcal{D}} d_{\mathcal{D}}(\rho_{\pi_{\mathrm{I}}}, \rho_{\pi_{\mathrm{E}}}),
\end{aligned}
$$

where $\|r\|_{\mathcal{D}} = \inf\{\sum_{i=1}^n |c_i| : r = \sum_{i=1}^n c_i D_i + c_0, \forall n \in \mathbb{N}, c_0, c_i \in \mathbb{R}, D_i \in \mathcal{D}\}$. Combining the above inequality with Lemma 2 completes the proof. □

## C   Analysis of Imitating-environments

We first introduce the error bound of policy evaluation without policy divergences, which will be used to prove Lemma 3 later.

**Lemma 7.** *Given an MDP with true transition model $M^*$, suppose the model error is $\epsilon_m$, i.e.,$\mathbb{E}_{(s,a)\sim\rho_{\pi_D}^{M^*}}\left[D_{\mathrm{KL}}\left(M^*(\cdot|s,a), M_\theta(\cdot|s,a)\right)\right] \leq \epsilon_m$ (see Eq. (3)), then for the data-collecting policy $\pi_D$ we have*

$$
\left| V_{\pi_D}^{M_\theta} - V_{\pi_D}^{M^*} \right| \leq \frac{\sqrt{2}R_{\max}\gamma}{(1-\gamma)^2}\sqrt{\epsilon_m}. \tag{16}
$$

*Proof.* The proof is similar to what we have done in Appendix A. First, we show that

$$
d_{\pi_D}^{M_\theta} = (1-\gamma)\sum_{t=0}^\infty \gamma^t \Pr(s_t = s; \pi_D, M_\theta, d_0) = (1-\gamma)(I - \gamma P_\theta)^{-1}d_0. \tag{17}
$$

where $P_\theta(s'|s) = \sum_{a\in\mathcal{A}} M_\theta(s'|s,a)\pi_D(a|s)$. Following the similar algebraic transformation in Lemma 4, we obtain that

$$
d_{\pi_D}^{M_\theta} - d_{\pi_D}^{M^*} = \gamma G(P_\theta - P^*)d_{\pi_D}^{M^*},
$$

where $G_\theta = (I - \gamma P_\theta)^{-1}$ and $G^* = (I - \gamma P^*)^{-1}$. Based on the Cauchy–Schwarz inequality, we have that

$$
D_{\mathrm{TV}}(d_{\pi_D}^{M_\theta}, d_{\pi_D}^{M^*}) = \frac{\gamma}{2}\|G_\theta(P_\theta - P^*)d^{M^*}\|_1 \leq \frac{\gamma}{2}\|G_\theta\|_1\|(P_\theta - P^*)d_{\pi_D}^{M^*}\|_1.
$$

We first show that $\|G_\theta\|_1$ is bounded as

$$
\|G_\theta\|_1 = \|\sum_{t=0}^\infty \gamma^t P_\theta^t\|_1 \leq \sum_{t=0}^\infty \gamma^t \|P_\theta\|_1^t \leq \sum_{t=0}^\infty \gamma^t = \frac{1}{1-\gamma}.
$$

We then show that $\left\|(P_\theta - P^*)d_{\pi_D}^{M^*}\right\|_1$ is bounded,

$$
\begin{aligned}
\left\|(P_\theta - P^*)d_{\pi_D}^{M^*}\right\|_1 &\leq \sum_{s',s} |P_\theta(s'|s) - P^*(s'|s)|d_{\pi_D}^{M^*}(s) \\
&\leq \sum_{s',s,a} |M_\theta(s'|s,a) - M^*(s'|s,a)|\pi_D(a|s)d_{\pi_D}^{M^*}(s) \\
&= 2\mathbb{E}_{(s,a)\sim\rho_{\pi_D}^{M^*}}[D_{\mathrm{TV}}(M_\theta(\cdot|s,a), M^*(\cdot|s,a))].
\end{aligned}
$$

Thanks to Pinsker's inequality and Jensen's inequality, we can get that

$$
D_{\mathrm{TV}}(d_{\pi_D}^{M_\theta}, d_{\pi_D}^{M^*}) \leq \frac{\sqrt{2}\gamma}{2(1-\gamma)}\sqrt{\epsilon_m}. \tag{18}
$$

From Lemma 6, we obtain that

$$\left| V_{\pi_D}^{M_\theta} - V_{\pi_D}^{M^*} \right| \leq \frac{R_{\max}}{1-\gamma} \sum_{(s,a)} \left| \rho_{\pi_D}^{M_\theta}(s,a) - \rho_{\pi_D}^{M^*}(s,a) \right|$$

$$\leq \frac{R_{\max}}{1-\gamma} \sum_s \left| d_{\pi_D}^{M_\theta}(s) - d_{\pi_D}^{M^*}(s) \right| \sum_a \pi_D(a|s)$$

$$\leq \frac{\sqrt{2}R_{\max}\gamma}{(1-\gamma)^2} \sqrt{\epsilon_m},$$

which concludes the proof. □

## C.1   Proof of Lemma 3

*Proof of Lemma 3.* Derived by the triangle inequality, the evaluation error can be decomposed into three parts.

$$|V_\pi^{M^*} - V_\pi^{M_\theta}| \leq |V_\pi^{M^*} - V_{\pi_D}^{M^*}| + |V_{\pi_D}^{M^*} - V_{\pi_D}^{M_\theta}| + |V_{\pi_D}^{M_\theta} - V_\pi^{M_\theta}|.$$

For the second term on the RHS, according to Lemma 7, we have

$$|V_{\pi_D}^{M^*} - V_{\pi_D}^{M_\theta}| \leq \frac{\sqrt{2}R_{\max}\gamma}{(1-\gamma)^2} \sqrt{\epsilon_m}.$$

For the first term, applying Lemma 5 and Lemma 6, we get that

$$|V_\pi^{M^*} - V_{\pi_D}^{M^*}| \leq \frac{2R_{\max}}{1-\gamma} D_{\mathrm{TV}}(\rho_\pi^{M^*}, \rho_{\pi_D}^{M^*})$$

$$\leq \frac{2R_{\max}}{(1-\gamma)^2} \mathbb{E}_{s \sim d_{\pi_D}^{M^*}} [D_{\mathrm{TV}}(\pi(\cdot|s), \pi_D(\cdot|s))]$$

$$\leq \frac{\sqrt{2}R_{\max}}{(1-\gamma)^2} \sqrt{\epsilon_\pi}.$$

Similar results hold for the third term, meaning that $|V_{\pi_D}^{M_\theta} - V_\pi^{M_\theta}| \leq \frac{\sqrt{2}R_{\max}}{(1-\gamma)^2} \sqrt{\epsilon_\pi}$. Combining the above three bounds completes the proof. □

## C.2   Proof of Theorem 3

*Proof of Theorem 3.* Due to Lemma 6, we obtain that

$$|V_\pi^M - V_\pi^{M^*}| \leq \frac{2R_{\max}}{1-\gamma} D_{\mathrm{TV}}(\rho_\pi^M, \rho_\pi^{M^*})$$

$$\leq \frac{2R_{\max}}{1-\gamma} (D_{\mathrm{TV}}(\rho_\pi^M, \rho_{\pi_D}^M) + D_{\mathrm{TV}}(\rho_{\pi_D}^M, \rho_{\pi_D}^{M^*}) + D_{\mathrm{TV}}(\rho_{\pi_D}^{M^*}, \rho_\pi^{M^*})).$$

The last inequality follows the triangle inequality. For the term $D_{\mathrm{TV}}(\rho_{\pi_D}^M, \rho_{\pi_D}^{M^*})$, we obtain that

$$D_{\mathrm{TV}}(\rho_{\pi_D}^M, \rho_{\pi_D}^{M^*}) = \frac{1}{2} \sum_{s,a} \left| \sum_{s'} \left( \mu^M(s,a,s') - \mu^{M^*}(s,a,s') \right) \right|$$

$$\leq \frac{1}{2} \sum_{s,a,s'} \left| \mu^M(s,a,s') - \mu^{M^*}(s,a,s') \right|$$

$$= D_{\mathrm{TV}}(\mu^M, \mu^{M^*}).$$

From Eq.(8), we derive that

$$D_{\mathrm{TV}}(\rho_{\pi_D}^M, \rho_{\pi_D}^{M^*}) \leq \sqrt{2D_{\mathrm{JS}}(\rho_{\pi_D}^M, \rho_{\pi_D}^{M^*})} \leq \sqrt{2D_{\mathrm{JS}}(\mu^M, \mu^{M^*})}.$$

Derived by Lemma 5, for the first term $D_{\mathrm{TV}}(\rho_\pi^M, \rho_{\pi_D}^M)$, we get that

$$D_{\mathrm{TV}}(\rho_\pi^M, \rho_{\pi_D}^M) \leq \frac{1}{1-\gamma} \mathbb{E}_{s \sim d_\pi^M} \left[ D_{\mathrm{TV}}\big(\pi(\cdot|s), \pi_D(\cdot|s)\big) \right]$$

$$\leq \frac{\sqrt{2}}{2(1-\gamma)} \mathbb{E}_{s \sim d_\pi^M} \left[ \sqrt{D_{\mathrm{KL}}\big(\pi(\cdot|s), \pi_D(\cdot|s)\big)} \right].$$

$$\leq \frac{\sqrt{2}}{2(1-\gamma)} \sqrt{\epsilon_\pi}.$$

The last two inequalities follow Pinsker's inequality and the definition of $\epsilon_\pi$ respectively. Similarly, for the second term $D_{\mathrm{TV}}(\rho_{\pi_D}^{M^*}, \rho_\pi^{M^*})$, we have that

$$D_{\mathrm{TV}}(\rho_{\pi_D}^{M^*}, \rho_\pi^{M^*}) \leq \frac{\sqrt{2}}{2(1-\gamma)} \sqrt{\epsilon_\pi}.$$

Combining the above three upper bounds completes the proof. $\qquad\square$

### C.3 Environment-learning with GAIL

---
**Algorithm 1** Environment-learning with GAIL

---
1: **Input:** data-collecting policy $\pi_D$, total iterations $N$, model update iteration $N_G$, discriminator update iteration $N_D$.
2: Initialize discriminator $D$, model $M_\theta$, and empty dataset $\mathcal{B}^*$ as well as $\mathcal{B}$.
3: $\mathcal{B}^* \leftarrow$ Collect samples using $\pi_D$ in model $M^*$.
4: **for** $N$ iterations **do**
5:     **for** $N_G$ iterations **do**
6:         $\mathcal{B} \leftarrow$ Collect samples using $\pi_D$ in model $M_\theta$.
7:         Assign rewards to state-action-next-state pairs in $\mathcal{B}$ by discriminator $D$.
8:         Update model $M_\theta$ by maximizing rewards with samples from $\mathcal{B}$.
9:     **end for**
10:     **for** $N_D$ iterations **do**
11:         Update discriminator $D$ by maximizing the following function:

$$\sum_{(s,a,s') \in \mathcal{B}} [\log(D(s,a,s'))] + \sum_{(s,a,s') \in \mathcal{B}^*} [\log(1 - D(s,a,s'))].$$

12:     **end for**
13: **end for**
14: **Output:** environment model $M_\theta$.

---

The process of applying GAIL to learn the environment transition model is summarized in Algorithm 1.

## D  Wasserstein GAIL

Similar to Wasserstein GAN (WGAN) [4], we can also introduce Wasserstein distance into GAIL. We call such an algorithm as Wasserstein GAIL (WGAIL for short). Specifically, the discriminator is selected from all 1-Lipschitz function classes by considering the following optimization problem.

$$\max_{D \in ||\mathcal{D}||_{\mathrm{Lip}} \leq 1} \mathbb{E}_{(s,a) \sim \rho_{\pi_{\mathrm{E}}}} [D(s,a)] - \mathbb{E}_{(s,a) \sim \rho_\pi} [D(s,a)]$$

Due to computation intractability, we cannot compute all 1-Lipschitz functions in practice, and thus we are shifted to its neural network approximation, where $D$ is parameterized by certain neural networks. As our result suggests, this method can still generalize well when its model complexity is controlled. However, ordinary neural networks are often not Lipschitz continuous. To maintain a good approximation to 1-Lipschitz continuous function classes, the gradient penalty technique is

**Algorithm 2** Wasserstein GAIL
---
1: **Input:** Expert demonstrations $\mathcal{B}^*$, total iterations $N$, policy update iterations $N_G$, discriminator update iterations $N_D$.
2: Initialize discriminator $D$, policy $\pi$, and an empty dataset $\mathcal{B}$.
3: **for** $N$ iterations **do**
4:     **for** $N_G$ iterations **do**
5:         $\mathcal{B} \leftarrow$ Collect samples using policy $\pi$.
6:         Assign scaled rewards to state-action pairs in $\mathcal{B}$ by discriminator $D$.
7:         Update policy $\pi$ by maximizing rewards with samples from $\mathcal{B}$.
8:     **end for**
9:     **for** $N_D$ iterations **do**
10:        Update discriminator $D$ by maximizing Eq. (19) with samples from $\mathcal{B}^*$ and $\mathcal{B}$.
11:     **end for**
12: **end for**
13: **Output:** policy $\pi$.
---

Table 2: Information about tasks in imitating policies.

| Tasks | State Dimension | Action Dimension | Episode Length |
|---|---|---|---|
| HalfCheetah-v2 | 17 | 6 | 1000 |
| Hopper-v2 | 11 | 3 | 1000 |
| Walker2d-v2 | 17 | 6 | 1000 |

introduced in WGAN [21]. This technique adds a regularization term that employs a quadratic cost to the gradient norm. Hence, denoting $(s, a)$ as $z$, the loss function for the discriminator in WGAIL is:

$$L(D) = \mathbb{E}_{z \sim \rho_\pi} \big[ D(z) \big] - \mathbb{E}_{z \sim \rho_{\pi_{\text{E}}}} \big[ D(z) \big] + \lambda \mathbb{E}_{z \sim \tilde{\rho}} \big[ \big( \|\nabla_z D(z)\| - 1 \big)^2 \big], \tag{19}$$

where $\tilde{\rho}$ is a mixing distribution of $\rho_\pi$ and $\rho_{\pi_{\text{E}}}$, and $\lambda$ is a positive regularization coefficient ($\lambda = 10$ performs well in practice). Following [24], we also scale reward function (discriminator's output) properly to stabilize training. This is important because the optimization in WGAIL is different from the one in WGAN. Concretely, reinforcement learning algorithms often use the evaluation value rather than the gradient information to perform gradient descent. Without scaling, rewards given by the discriminator often fluctuate, which may lead to an unstable optimization. To tackle this issue, at each iteration, we firstly centralize the given rewards by subtracting the mean and subsequently scale them by dividing the range (the difference between the maximal value the minimal value). The algorithm procedure is outlined in Algorithm 2.

# E  Experiment Details

## E.1  Imitating Policies

We evaluate the considered algorithms on OpenAI Gym [10] benchmark tasks. Information about state dimension, action dimension, and episode length information is listed in Table 2. We run the state-of-the-art algorithm SAC [22] for 1 million samples to obtain expert policies. All imitation learning approaches use 2-layer MLP policy network with 100 hidden sizes and $tanh$ activation function. Except for DAgger that continues to collect new samples and query expert policies (i.e., DAgger collects 1000 samples and gets action labels from expert policies per 5000 iterations), all methods are provided the same 3 expert trajectories with length 1000. Key parameters of BC and DAgger are give in Table 3 and Table 4, respectively. Other methods including GAIL, FEM [1] and GTAL [49] use TRPO [41] to optimize policies, and key parameters are given in Table 5. All experiments run with 3 random seeds (namely, 100, 200 and 300). During the training process, we periodically evaluate the learned policies on true environments with 10 trajectories. Learning curves are given in Figure 7. The final performance of imitated policies and expert policies are listed in Table 6. Please refer to our source code in supplementary materials for other details.

Table 3: Key parameters of Behavioral Cloning.

| Parameter | Value |
|---|---|
| learning rate | 3e-4 |
| batch size | 128 |
| total number of iters | 100k |

Table 4: Key parameters of DAgger.

| Parameter | Value |
|---|---|
| learning rate | 3e-4 |
| batch size | 128 |
| number of total training iterations | 100k |
| collecting frequency | 5k |
| number of new demonstrations per iteration | 1k |

Table 5: Key parameters of GAIL, AIRL, FEM and GTAL.

| Parameter | Value |
|---|---|
| number of generator iterations | 5 |
| number of discriminator iterations | 1 |
| number of rollout samples per iteration | 1k |
| total number of collecting samples | 3M |
| maximal KL divergence | 0.01 |

Table 6: Discounted returns of learned policies. We use $\pm$ to denote the standard deviation

| | Tasks | Expert | BC | DAgger | FEM | GTAL | GAIL | WGAIL | AIRL |
|---|---|---|---|---|---|---|---|---|---|
| $\gamma = 0.9$ | HalfCheetah-v2 | $10.59 \pm 0.00$ | $4.02 \pm 0.97$ | $8.06 \pm 0.33$ | $-11.40 \pm 1.67$ | $-14.39 \pm 4.37$ | $1.86 \pm 0.08$ | $-2.13 \pm 2.78$ | $1.57 \pm 1.78$ |
| | Hopper-v2 | $10.85 \pm 0.00$ | $10.69 \pm 0.10$ | $10.86 \pm 0.02$ | $12.75 \pm 0.81$ | $13.93 \pm 0.74$ | $10.31 \pm 0.19$ | $11.30 \pm 1.06$ | $13.50 \pm 0.56$ |
| | Walker2d-v2 | $5.31 \pm 0.00$ | $5.60 \pm 0.20$ | $5.30 \pm 0.11$ | $9.86 \pm 0.87$ | $11.31 \pm 0.49$ | $5.24 \pm 0.41$ | $9.64 \pm 1.30$ | $5.96 \pm 0.56$ |
| $\gamma = 0.99$ | HalfCheetah-v2 | $511.99 \pm 0.00$ | $137.30 \pm 70.70$ | $465.37 \pm 4.56$ | $-148.64 \pm 10.50$ | $-193.40 \pm 88.66$ | $251.49 \pm 22.00$ | $315.21 \pm 57.95$ | $305.69 \pm 94.93$ |
| | Hopper-v2 | $275.81 \pm 0.00$ | $155.19 \pm 16.27$ | $276.10 \pm 0.15$ | $238.63 \pm 7.24$ | $227.96 \pm 18.18$ | $263.17 \pm 3.47$ | $178.17 \pm 106.70$ | $259.35 \pm 5.48$ |
| | Walker2d-v2 | $346.63 \pm 0.00$ | $244.67 \pm 30.28$ | $345.87 \pm 1.79$ | $129.21 \pm 16.38$ | $176.27 \pm 22.33$ | $338.60 \pm 9.28$ | $249.95 \pm 27.41$ | $314.11 \pm 34.82$ |
| $\gamma = 0.999$ | HalfCheetah-v2 | $4097.30 \pm 0.00$ | $536.68 \pm 384.66$ | $3730.81 \pm 31.57$ | $-1150.45 \pm 235.64$ | $-1509.39 \pm 877.41$ | $3338.52 \pm 191.06$ | $2670.44 \pm 437.00$ | $3303.42 \pm 262.79$ |
| | Hopper-v2 | $2223.49 \pm 0.00$ | $408.25 \pm 222.42$ | $1903.02 \pm 83.18$ | $1878.19 \pm 122.06$ | $1731.93 \pm 233.34$ | $2177.76 \pm 43.64$ | $1184.20 \pm 805.75$ | $2187.72 \pm 17.90$ |
| | Walker2d-v2 | $3151.77 \pm 0.00$ | $995.05 \pm 330.00$ | $2963.82 \pm 26.59$ | $1039.71 \pm 231.21$ | $1765.62 \pm 212.79$ | $2912.35 \pm 335.22$ | $1565.15 \pm 1006.01$ | $1877.53 \pm 651.77$ |

## E.2 Imitating Environments

To evaluate algorithms for imitating environments, we add necessary information (e.g., robot position information) to the original state space defined by OpenAI Gym [10]. This is important since we need the learned environment model to predict the position information, upon which we can compute rewards for policy evaluation in the learned environments. Followed prior works [31, 25], the true reward function is assumed to be known in advance. We also normalize the robot position information by dividing 10 (but the reward function is not normalized). We use SAC [22] to re-train a sub-optimal policy as what we have done when imitating policies. We collect samples using this sub-optimality on true environments. Algorithmic configuration for BC and GAIL is the same as the one of imitating policies. Different from imitating-policies, the model output space (action space) is not bounded between $-1$ and $+1$. To overcome this difficulty, we normalize the model's outputs with statistics obtained from given demonstrations. During the training process, we also periodically evaluate the policy value of data-collecting policies on the learned environment models.

Figure 7: Learning curves of imitation approaches ($\gamma = 0.999$) including DAgger, GAIL, AIRL, WGAIL, FEM, GTAL, and BC. The solid lines are mean of results and the shaded region corresponds to the stand deviation over 3 random seeds, while the dashed lines indicate the performance of expert policies.