[Reviews · NeurIPS 2020]

Review 1

Summary and Contributions: The paper provides upper bounds on the policy's value gap between the expert policy and the imitation policy as a function of the effective horizon. While naive imitation methods such as behavioral cloning show a policy value gap quadratic w.r.t the effective horizon, more advanced imitation algorithms such has GAIL show linear dependence at the horizon. The difference between the linear bound of GAIL and the quadratic bound of behavioral cloning lies in the fact that behavioral cloning can be viewed as minimizing the (KL) divergence between action distributions whereas GAIL directly minimizes the (Janson-Shanon) divergence between state-action occupancy measures. The authors also analyze the value gap in a model-based RL setup where expert demonstrations are used to learn a model of the environment. It is not surprising that by applying the same type of analysis, the authors revile that environment models trained with GAIL lead to lower policy evaluation errors than when trained with behavioral cloning. The authors also show that the policy evaluation error is comprised of two terms: one that depends on the model error, and one that depends on the divergence between the expert and said policy. The empirical evaluation shows that the performance of behavioral cloning decreases when the horizon is increased, while the performance of GAIL remains stable. Experiments are conducted with very few expert examples (3 trajectories in total). This evidence is consistent with the theoretical analysis.

Strengths: The paper is very rigorous.

Weaknesses: Although the analysis is clear about the advantages of GAIL over behavioral cloning, it is important to note that while the quadratic bound of behavioral cloning is correct, the linear bound of GAIL is probably approximately correct, I.e., its a PAC result.

Correctness: Yes. I believe that a factor of 2 is missing in the last line of the proof of lemma 8, but nothing significant beyond that.

Clarity: There are some weird phrasings, but overall the paper is well written and easy to follow.

Relation to Prior Work: There are few works that try to formalize the theoretical aspects, and in particular, the generalization characteristics of imitation learning algorithms. Therefore, the discussion about previous work is limited.

Reproducibility: Yes

Additional Feedback: It is possible that the authors mix between empirical and non-empirical Radmacher complexities? They describe Radmacher complexity but refer to it as empirical Radmacher.


Review 2

Summary and Contributions: The paper considered the differences between GAIL and BC for imitation learning. Theoretical bounds on the suboptimality of imitation policies are provided. The show that, because it approximates the state-action distribution rather than just action likelihoods, GAIL exhibits better scaling with horizon than BC. Empirical comparisons are shown that are consistent with this, and examples using GAIL or BC methods for learning an environment model.

Strengths: The main result of Theorem 4 in comparison to Theorem 1 provides an interesting perspective on these algorithms, and helps explain why GAIL might be preferable. I really like this result, and find it compelling. That Thm 1 is tight is important to support the conclusions drawn.

Weaknesses: One drawback of the results is that there is a dependence on the success of the optimisation for either BC / GAIL, which may in practice prove more important than the dependency on horizon. This seems inevitable in the theory, but could maybe be investigated experimentally. The experimental results are a relatively minor contribution. I would consider them as more of an illustration. The experimental results in 6.1 are meant to indicate that BC is less able to scale with gamma than GAIL. This appears to be the case, but that’s based on eyeballing the plots, not a rigorous analysis. The scope is quite limited too: we only have 3 different discount factors in each experiment. Moreover, we only see 3 domains, which share some similarity (all continuous control in MuJoCo) so it’s not the most conclusive study. Finally, some fairly arbitrary choices were made, e.g. to only use 3 expert trajectories. I’d be interested to see a little description of how they were chosen / how it affects the results. Results from model learning are quite ambiguous… it’s not clear what this experiment tells us?

Correctness: As far as I can tell the results are correct.

Clarity: I found the paper to be clear. The way the main results are contrasted makes the key message of the paper obvious to the reader.

Relation to Prior Work: Clear and well explained

Reproducibility: Yes

Additional Feedback: The shading on plots is unexplained. ===== I recommend acceptance of this work entirely on the basis of the theoretical result. I feel the empirical section could use work. My biggest issue is that in figure 1 we have to just eyeball plots to see if they seem to match the theoretical claims, and we therefore have no way to tell whether they are real or statistical noise, etc. Particularly with just 3 values of gamma, that's an issue. I hope the authors address this in the final version by doing a rigorous statistical analysis of whether the effect is present. The results of figure 3 still seem ambiguous overall, I'm not sure it adds much. Thank you for the clarification of the shading.


Review 3

Summary and Contributions: The paper presents theoretical analyses behavior cloning (BC) and generative adversarial IL (GAIL) for infinite horizon MDPs. The paper aims to bound the value function gap which is the gap between value functions of the expert policy and the imitating policy. The results indicate that, the value function gap of BC depends quadratically on the effective horizon, while the value function gap of GAIL depends linearly on the effective horizon. This implies that GAIL suffers less from the compounding error compared to BC. The paper also conducts similar analyses for model-based RL where BC and GAIL are used to learn a transition model. The results indicate similarly that GAIL suffers less from model biased compared to BC. Experiments on Mujoco tasks are conducted to support the theoretical results.

Strengths: - The paper presents thorough theoretical results of BC and GAIL under an infinite horizon setting. These results could help developing better methods. - The analysis for model-based RL is very interesting since it suggests that GAIL is a promising alternative to learn a transition model.

Weaknesses: - The experiments are conducted under a finite-horizon MDP setting, but the analysis is conducted under an infinite-horizon MDP setting. This inconsistency makes the experimental results quite weak at supporting the theoretical results.

Correctness: The claims based on the theoretical results appear to be correct. Though, I did not carefully check the proof in the supplementary materials.

Clarity: The paper is overall well written and describe most necessary details to understand the theoretical results (e.g., approximation errors and estimation errors). However, one issue is that the paper does not clearly describe the meaning of generalization which may cause ambiguity. This is because generalization in RL could mean either 1) generalization within the training MDP where we want the model to perform well on unseen state-action data points [1], and 2) generalization across MDPs where we want the model to perform well on unseen MDPs [2]. The former meaning is close to that in supervised learning, and I presume the paper considers this type of generalization. Still, I think the paper can be improved by clarifying the type of generalization the paper is considering. [1] R. Sutton. Generalization in Reinforcement Learning: Successful Examples Using Sparse Coarse Coding. NeurIPS 1996. [2] K. Cobbe, O. Klimov, C. Hesse, T Kim, J. Schulman. Quantifying Generalization in Reinforcement Learning. ICML 2019.

Relation to Prior Work: The paper well presents prior work and discuss the differences between the paper and the prior work.

Reproducibility: No

Additional Feedback: Overall, I enjoy reading the paper. It presents theoretical justifying the usage of GAIL over BC for infinite horizon problems. The results could also be used to develop new algorithms that have better theoretical performance than GAIL. Overall, I recommend acceptance. I have a comment/question that would like to the authors to address. - The paper conducts theoretical analyses under an infinite horizon MDP where the discount factor $\gamma$ determines the effective horizon. However, Mujoco tasks used in the experiments are (by default) finite horizon problems. Unless the Mujoco tasks are altered to be infinite horizon, I do not think that the experimental results can be used to support the theoretical results. ** Update after rebuttal ** I have read the rebuttal letter and other reviews. The authors clarify my concerns about the meaning of generalization and the experimental setting. I recommend accepting the paper.


Review 4

Summary and Contributions: The authors provide an analysis of behavior cloning and adversarial imitation learning to evaluate the value gap between expert and imitated policies as well as a learned environment model to the true model.

Strengths: The paper is generally well written and motivated and the proofs are clear. The technical and theoretical contributions seem strong and empirically validated.

Weaknesses: The approach used to compare BC seems out of date as there are many more recent approaches that address the compounding error problem and have been shown to achieve better results in imitation learning, such as maxent IRL and other more recent methods in IRL. Reactive policy matching can lead to bad states which is why value iteration and Q learning are used to estimate state-action pairs in terms of future cumulative rewards. For example, Max Ent IRL aims uses value iteration to capture feature preferences of the experts to model their behavior in unseen settings with the explicit goal of maximizing the distribution of the policy through near optimal state-action pairs rather than matching the expert policy directly. The learned policies may differ from expert demonstrations as long as the state-action features being optimized by the learner are similar to those of the demonstrator. In other words, there are multiple near-optimal policies that would be acceptable and are still indicative of the demonstrated behavior which are learned by this method reducing the effect of compounding errors and allowing for deviations from the expert state distribution. Please correct me if I am confused on this assumption on why [35], as well as [1] and [46] as proposed in [21], are used instead of these other approaches. I am open to being convinced that the comparisons provided in the paper are sufficient but I currently do not feel this way. Figures should be enlarged. They are not clear. The paper feels incomplete. The authors mention "... previous MBRL methods mostly involve a BC-like transition learning component, which is possible to be replaced by a GAIL-style learner to improve the generalization ability" an examination of this would be a nice addition to the paper. Essentially the authors analyze these two approaches to imitation learning and discover different bounds on their performance but do not build on what was presented.

Correctness: Yes

Clarity: Yes

Relation to Prior Work: Yes, but I would prefer a stronger comparison to other approaches in imitation learning that seem to be more robust to the errors described here.

Reproducibility: Yes

Additional Feedback:

[Author Response · NeurIPS 2020]

**Response to Reviewer 1** Thanks for your useful suggestion. We will add more discussion about the generalization
bounds of other imitation learning methods and fix the mistake in the proof of Lemma 8 in the future version.

**C1**: We have provided a PAC result for BC in Corollary 10 in the Appendix for a direct comparison. Please note that
the PAC bound is based on sample-approximation, hence the dependency on the effective horizon does not change.

**C2:** Rademacher complexity in this paper refers to its empirical version and we will clarify this in the future version.

**Response to Reviewer 2** Thanks for your insightful comments.

**Q1:** The choice of 3 expert trajectories and its effect?

**A1:** We choose 3 trajectories to better demonstrate that when the estimation error in PAC bounds is large, the $\gamma$-
dependency of GAIL is better than BC. Consistent with prior works [20, 27], we do observe that when more expert
trajectories are provided (e.g., 10 trajectories), BC achieves comparable performance to GAIL since the estimation
errors for both methods are very small and the $\gamma$-dependency does not dominate.

**Q2:** Results of model learning experiment?

**A2:** We use policy evaluation errors to evaluate the quality of model learning. The results in Section 6.2 indicate that
the environment recovered by GAIL could be better.

**C1:** The shading on plots refers to the standard deviation over 3 random seeds. We will clarify this in the future version.

**Response to Reviewer 3** Thanks for your helpful advice. The word "generation" means that the policy could perform
well on unseen states in the training MDP, and we will clarify this in the future version.

**Q1:** The difference regarding the horizon settings between theoretical analysis and experiments.

**A1**: We cannot run infinite steps in an infinite-horizon MDP if there are no absorbing states. Therefore, we choose to
use a finite horizon MDP with a large horizon $H = 1/(1 - \gamma)$, such that there is only a small difference to the infinite
horizon. To see this, assuming the reward function is bounded within $[0, 1]$, we have $|\sum_{t=H}^{\infty} \gamma^t r(s_t, a_t)| \leq \epsilon \Longrightarrow H \geq$
$\frac{1}{1-\gamma} \log(\frac{1}{(1-\gamma)\epsilon})$.

**Others:** For your choice of "No" to the reproducibility evaluation, we would like to point out that the proof, source
code and relevant dataset are provided in the supplementary material to help reproduce the work.

**Response to Reviewer 4** We thank you for your insightful suggestion. We will enlarge the picture in the future version.

**Q1:** The assumption on why [35], as well as [1] and [46] as proposed in [21], are used instead of these other approaches
(MaxEnt IRL)? The comparisons provided in the paper are sufficient?

**A1:** Since our main contribution is the theoretical results, the conducted experiments are used to verify the theory. Note
that GAIL [20] is closely related to MaxEnt IRL by incorporating the $\gamma$-discounted causal entropy (see Equation (1) in
[20]). To remedy your concern, here we provide an additional experiment that considers the SOTA MaxEnt IRL method
called AIRL [Fu et al., ICLR 2018], which will be included in the paper. We would like to consider the error bounds of
MaxEnt IRL in the future.

Compared to prior works [20, 26, 27], we have additionally considered the BC-like method DAgger [39] and we think
the empirical comparisons are sufficient, though the empirical comparisons are not the main scope of this paper.

Figure 1: Performance on Hopper-v2.

| $\gamma$ | Expert | BC | GAIL | AIRL |
|---|---|---|---|---|
| 0.9 | $10.85 \pm 0.00$ | $10.69 \pm 0.10$ | $10.31 \pm 0.19$ | $13.11 \pm 0.65$ |
| 0.99 | $275.81 \pm 0.00$ | $155.19 \pm 16.27$ | $263.17 \pm 3.47$ | $258.96 \pm 4.50$ |
| 0.999 | $2223.49 \pm 0.00$ | $408.25 \pm 222.42$ | $2177.76 \pm 43.64$ | $2087.75 \pm 154.99$ |

Table 1: Discounted returns of learned policies on Hopper-v2 (performance of other
methods can be found in the original Appendix). We use $\pm$ to denote the standard
deviation.

**Q2:** The paper feels incomplete. The authors mention "...".

**A2:** The main contribution of this paper is to provide error bounds on imitating policies and environments and to give
insights for future algorithm designs. For MBRL, our theoretical results suggest that environment-learning by GAIL
could be better. Combining the environment-learning with all kinds of policy optimization algorithms, while is very
interesting, is beyond the main scope of this paper and is left for future work.

[Meta-Review · NeurIPS 2020]

The reviewers appreciated the rebuttal. In the discussion everybody agreed that the theoretical contribution is strong (which on its own warrants the acceptance of the paper) while the experimental section could still be improved quite a bit.